# Ophiuchus: Incentivizing Tool-augmented "Think with Images" for Joint Medical Segmentation, Understanding and Reasoning

Yankai Jiang [* 1 4]   Yujie Zhang [* 2 3]   Peng Zhang [1]   Wenjie Li [2 5]   Yichen Li [6]
Jintai Chen [7]   Xiaoming Shi [8]   Shihui Zhen [1]

## Abstract

Recent medical MLLMs have made significant progress in generating step-by-step textual reasoning chains. However, they still struggle with complex clinical tasks that necessitate dynamic and iterative focusing on fine-grained visual regions. To close this gap, we introduce **Ophiuchus**, a versatile, tool-augmented framework that equips an MLLM to (i) decide when fine-grained visual evidence is needed, (ii) determine where to probe and ground within the medical image, and (iii) seamlessly weave the relevant sub-image content back into an interleaved, multimodal chain of thought for precise segmentation and diagnosis. **Ophiuchus** moves beyond mere tool-calling by tightly fusing the MLLM's inherent grounding and reasoning capabilities with external tools, enabling more accurate and trustworthy decisions. The core of our method is a three-stage training strategy: cold-start SFT for basic tool selection; self-reflection fine-tuning to strengthen decision revision; and agentic tool reinforcement learning to elicit sophisticated, expert-like diagnostic behaviors. Extensive experiments show that **Ophiuchus** consistently outperforms both closed-source and open-source SOTA methods across diverse medical benchmarks, including VQA, detection, and reasoning-based segmentation. Our project code is available at https://github.com/SII-zyj/Ophiuchus.

[*]Equal contribution  [1]Zhejiang University [2]Shanghai Innovation Institute [3]Shanghai Key Lab of Intelligent Information Processing, College of Computer Science and Artificial Intelligence, Fudan University [4]Shanghai Artificial Intelligence Laboratory [5]Department of Orthopaedics, Ruijin Hospital, College of Health Science and Technology, Shanghai Jiao Tong University School of Medicine [6]Huazhong University of Science and Technology [7]Information Hub, HKUST (Guangzhou) [8]East China Normal University. Correspondence to: Shihui Zhen <11718287@zju.edu.cn>.

*Proceedings of the 43$^{rd}$ International Conference on Machine Learning*, Seoul, South Korea. PMLR 306, 2026. Copyright 2026 by the author(s).

## 1. Introduction

Multimodal Large Language Models (MLLMs) have exhibited remarkable performance across a wide range of medical image analysis tasks such as visual question answering (VQA) (Li et al., 2023; Chen et al., 2024a), disease diagnosis (Sun et al., 2025b; Liu & Song, 2025), segmentation (Bai et al., 2024; Wang et al., 2025a), and report generation (Bassi et al., 2025; Wang et al., 2025b). Recent breakthroughs in chain-of-thought (CoT) techniques (Wei et al., 2022) have further advanced the reasoning capabilities of medical MLLMs. Numerous studies employ supervised fine-tuning (SFT) (Sun et al., 2025a) or reinforcement learning (RL) (Lai et al., 2025b; Su et al., 2025b) to push these models beyond the direct-prediction paradigm, enabling step-by-step reasoning to address complex clinical challenges and improve diagnostic decision support.

Despite these advancements, current medical MLLMs still show critical limitations in how they interact with visual information during reasoning processes. First, they often attend to irrelevant regions while missing fine-grained evidence, such as tiny structures, intricate abnormalities, subtle lesion cues, and nuanced pathological semantics (Wang et al., 2025a). Such omissions severely degrade diagnostic performance. In essence, these weaknesses stem from a static, global-perception paradigm: the models primarily rely on image-level representations and lack the key ability to actively and adaptively probe and explore localized, fine-grained visual details. Moreover, prevailing medical MLLMs express intermediate reasoning steps exclusively in text and lack a "look-again" mechanism during thinking, leading to the loss of critical visual information. Ideally, MLLMs should autonomously interact with task-relevant image regions in a dynamic, iterative manner, revisiting and revising predictions as new visual evidence emerges to improve perceptual decision-making, as shown in Figure 1.

The aforementioned challenges motivate a fundamental rethinking of how medical MLLMs engage more seamlessly with fine-grained visual information during reasoning. This leads us to propose **Ophiuchus**, a novel and versatile framework capable of interleaved vision-language reasoning through thinking with tools across diverse med-

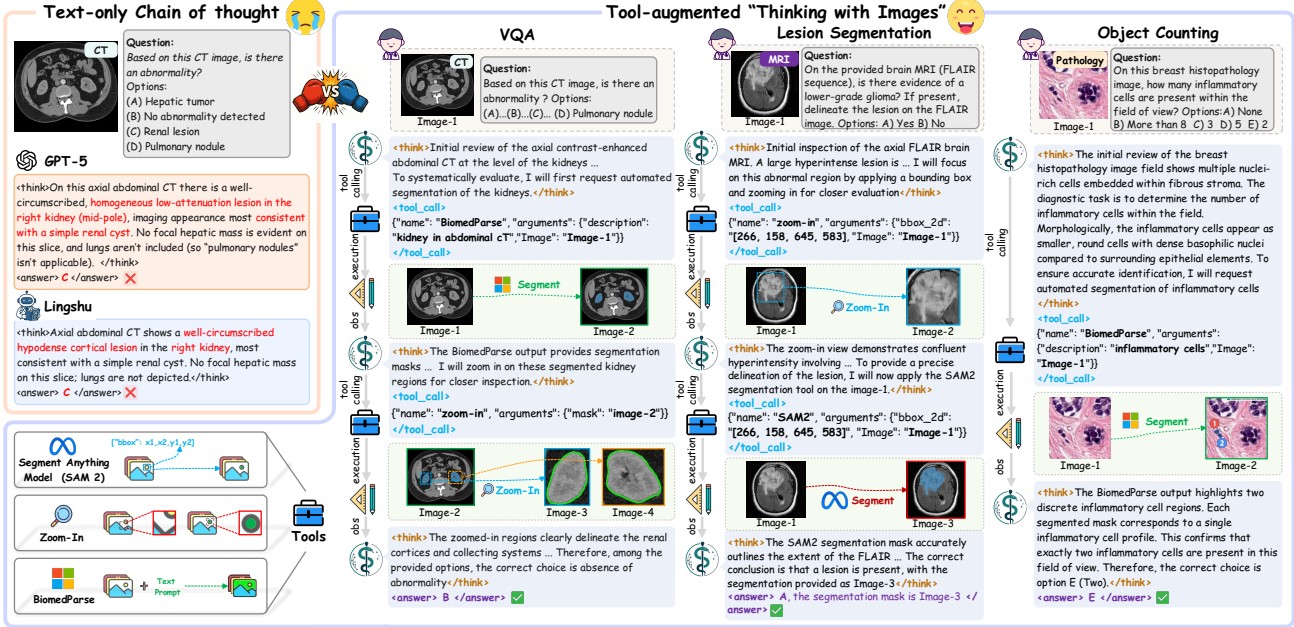

Figure 1. **Overview of the tool-augmented "thinking with images" paradigm.** Compared with (a) text-only CoT, which overlooks fine-grained, task-critical regions and thus limits understanding and increases hallucinations and false positives, (b) our tool-augmented interleaved vision-language reasoning adaptively plans tool invocations, inspects local regions, and incorporates the evidence into subsequent reasoning, yielding more accurate diagnostic cognition across diverse medical tasks.

ical tasks, including VQA, detection, and segmentation. **Ophiuchus** can decide whether to actively integrate external visual tools (*e.g.*, SAM2 (Ravi et al., 2024), Biomed-Parse (Zhao et al., 2024), and zoom-in) directly into the reasoning loop to iteratively manipulate and interpret key visual content. This paradigm shift from pure text-based reasoning to a more grounded and interpretable tool-augmented visual cognition—a new frontier where reasoning is continuously intertwined with ongoing visual perception.

To develop **Ophiuchus**, we construct a high-quality dataset of 64k samples with annotations for grounding key visual regions and explicit tool-invocation trajectories, and we propose a three-stage training protocol (Figure 2). First, we introduce cold-start SFT with tool-integrated reasoning data, enabling the model to learn how to invoke appropriate external vision tools and extract useful information from their outputs to solve questions beyond its internal capabilities. Second, we conduct fine-tuning with a new self-reflection sampling (SRS) strategy that further strengthens self-correction, enabling the model to assess whether its tool choices meet expectations based on intermediate results and to adjust accordingly. Finally, we introduce agentic tool reinforcement learning (ATRL) with fine-grained rewards that further strengthens reasoning, enabling the model to autonomously discover effective tool-use policies rather than merely imitating the static tool-use trajectories from the SFT stage. We conduct a comprehensive evaluation on 8 benchmarks, spanning both in-domain and zero-shot sce-

narios, to rigorously assess the performance of **Ophiuchus**. Results show that our model exhibits strong interleaved vision-language reasoning with tools and significantly exceeds the SOTA MLLMs across all tasks. Our contributions can be summarized as follows:

- We propose tool-augmented "think with images", an interleaved multimodal reasoning paradigm that enables medical MLLMs to adaptively integrate localized visual evidence into the evolving reasoning chain.

- We develop a principled, multi-stage training framework that cultivates tool-augmented reasoning: the model progresses from early tool exploration to advanced orchestration, enabling it to solve complex diagnostic problems that require detailed visual inspection.

- **Ophiuchus** establishes new state-of-the-art performance across multiple medical benchmarks. More importantly, it demonstrates an evolution in reasoning: from simple tool-calling to thinking with tools.

## 2. Related Works

**Medical MLLMs for Fine-Grained Vision Understanding.** Multimodal large language models have demonstrated promising performance in handling a wide range of medical imaging modalities and tasks (Li et al., 2023; Chen et al., 2024a; Yang et al., 2024; Lin et al., 2025). To enable region-level perception and assessment, numerous ap-

proaches (Chen et al., 2024b; Xie et al., 2024; Shui et al., 2025; Deng et al., 2025) introduce region-of-interest (ROI) supervision, teaching medical MLLMs to ground critical regions. Moreover, some studies (Bai et al., 2024; Wang et al., 2025a; Huang et al., 2025) have also explored pixel understanding in MLLMs by combining these models with segmentation tasks. These methods require MLLMs to learn an implicit $[seg]$ token and involve additional fine-tuning with a separate pixel decoder. Despite direct fine-grained supervision, these models still exhibit limited grounding accuracy and mask precision (Zhu et al., 2025b; Nath et al., 2025), and they often underperform task-specific models. In contrast, we train the MLLM to act as a competent agent that effectively invokes external tools across diverse queries, enabling it to surpass inherent limitations in pixel-level grounding and to solve tasks beyond its native capacity.

**Tool-Augmented Medical Agents.** Augmenting medical MLLMs with external tools is an active direction that extends capabilities beyond standalone models by leveraging dedicated functions or expert models for quantitative assessment. MMedAgent (Li et al., 2024) curates an instruction-tuning corpus covering six medical tools, enabling task-specific tool invocation and result aggregation. Similarly, VILA-M3 (Nath et al., 2025) trains models to trigger medical expert models for tasks such as segmentation and classification. AURA (Fathi et al., 2025) composes specialized medical models as tools, e.g., MedSAM (Ma et al., 2024) for segmentation and CheXAgent (Chen et al., 2024c) for VQA. However, these methods remain restricted to fixed perception behaviors due to fragmented reasoning composed of disconnected tool invocations that undermine coherence and holistic planning. MedAgent-Pro (Wang et al., 2025c) proposes a hierarchical multi-agent framework that combines GPT-4o (Achiam et al., 2023) with retrieval and analysis tools for disease diagnosis. Although these approaches support sequential composition of multiple tools, their reliance on predefined workflows limits generalization and blocks recovery from tool failures. In contrast, **Ophiuchus** achieves tool-augmented "think with images" through a three-stage training framework, enabling seamless orchestration of diverse tools and iterative reasoning to reflect on tool outputs.

## 3. Methods

We first present an overview of **Ophiuchus**. Then we introduce our three-stage training framework including cold-start SFT, self-reflection fine-tuning, and agentic tool reinforcement learning. Finally, we introduce our dataset curation.

### 3.1. Ophiuchus

**Ophiuchus** is a unified multimodal agent that can "think with images" by adaptively invoking and sequencing tools throughout CoT reasoning. As illustrated in Figure 1, given a user question $Q$ and an input image $I$, **Ophiuchus** gener-

ates a multi-step reasoning path $R$ to derive the final answer. This reasoning path $R$ can be represented as an $N$-step chain: $R = \{(r_n, t_n, o_n)\}_{n=1}^{N}$, where each step comprises natural-language thoughts $r_n$, a tool invocation $t_n$ for further image inspection, and the resulting observation $o_n$. The iterative thought-tool-observation loop continues until the model reaches a conclusive answer in the final reasoning thoughts $r_N$ or when predefined limits on interaction turns are reached. The core components are detailed below.

**(1) Available Tools for Image Analysis.** We equip **Ophiuchus** with three useful tools, including:

• SAM2 (Ravi et al., 2024). We employ SAM2 as one of the segmentation tools due to its superior performance and efficient inference speed. Leveraging the bounding boxes provided by **Ophiuchus**, SAM2 can generate precise, fine-grained mask for the target object.

• BiomedParse (Zhao et al., 2024), which takes an image and a text prompt as inputs. The text prompt specifies the object type for segmentation, then BiomedParse outputs a segmentation mask. Unlike SAM2, the text-driven Biomed-Parse enables complete delegation of region localization to the tool, whereas SAM2 still relies on the MLLM to produce bounding boxes (bboxes) as prompts. Although both support segmentation, BiomedParse complements rather than duplicates SAM2; providing an effective alternative when the MLLM cannot reliably generate bboxes (Appendix D.2).

• An image zoom-in function, which accepts an image together with either bboxes or masks as input. With bboxes, the function returns zoom-in crops of the specified regions; with masks, it returns the corresponding zoom-in crops and additionally delineates the target object contours (making boundary morphology explicit in this way facilitates recognition of lesion-specific shape cues).

**(2) Tool-Integrated Reasoning.** At the $n$-th reasoning step, **Ophiuchus** $\mathcal{M}$ can autonomously decide, after textual CoT thoughts $r_n$, whether to directly produce an answer, or invoke a tool $t_n$ for further image inspection. The tool's outputs, such as masks or cropped images, are sequentially indexed to maintain order and appended to the ongoing trajectory, allowing the model to reason over all previous context. This reasoning process can be formulated as: $(r_n, t_n) = \mathcal{M}(Q, I, R_{<n})$, where $R_{<n} = \{(r_i, t_i, o_i)\}_{i<n}$ denotes the reasoning history before step $n$. Each tool invocation $t_i$ can target either the original input image or any previous observation along the trajectory via explicit indices. Our design enables the MLLM to learn dynamic tool composition across diverse tasks, coupling visual cues with textual reasoning for accurate decision-making.

### 3.2. Training Framework

Our training procedure consists of three stages (Figure 2): (1) cold-start SFT to establish basic tool-invocation and

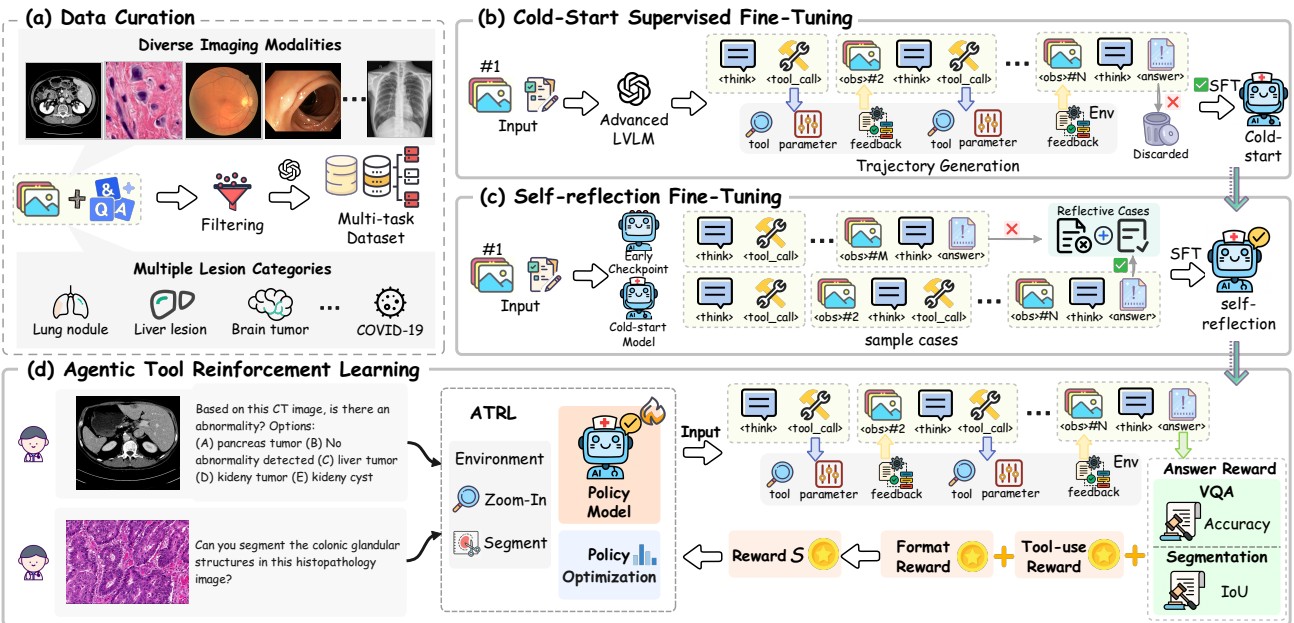

*Figure 2.* **Overview of the Ophiuchus.** A versatile "think with images" framework that integrates zoom and segmentation tools, trained through cold-start SFT, self-reflection fine-tuning and ATRL.

visual-interaction capabilities. (2) self-reflection fine-tuning that fosters self-correction behaviors, enabling the model to reassess tool outputs and, when necessary, to revise its tool-invocation decisions. (3) ATRL. We train with RL and carefully designed fine-grained rewards to further incentivize reasoning, driving tool use from early exploration to efficient, accurate exploitation and orchestration via trial-and-error interactions across diverse tasks.

**Cold-Start Supervised Fine-Tuning.** Prompting alone is insufficient for multimodal agents to reliably select and invoke tools in real-world medical tasks. We therefore bootstrap **Ophiuchus** with supervised fine-tuning on a cold-start set $\mathcal{D}_{\text{cold}}$ that provides ground-truth reasoning and tool-call trajectories. We train by maximizing the likelihood of the interleaved reasoning and tool-invocation tokens conditioned on the image and question. The formula of this objective is given in Appendix B.

**Self-Reflection Fine-Tuning.** Moreover, considering the complexity of real-world medical tasks, errors can occur even when tools are invoked. It is therefore crucial for the model to exhibit reflective behavior over tool outputs. To this end, we introduce a novel self-reflection sampling (**SRS**) strategy that further strengthens the model's self-correction capability. Specifically, we sample self-reflective cases from the model's training dynamics. For challenging questions, when the agent moves from an incorrect answer at an early checkpoint to a correct answer at a later checkpoint, and its tool sequence differs between the two, we regard this as evidence of self-reflective strategy revision. Formally, we define a self-reflective reasoning process as a pair of trajec-

tories $(R_{\text{early}}, R_{\text{late}})$ for the same input $(I, Q)$ observed at different checkpoints, where the tool-invocation sequences differ and the answer correctness flips:

$$\exists\,(u,v): (t^u_{\text{early}} \neq t^v_{\text{late}}) \,\wedge\, \text{Correct}(R_{\text{early}}) = 0$$
$$\wedge\; \text{Correct}(R_{\text{late}}) = 1, \tag{1}$$

where $u$ and $v$ denote arbitrary steps in two different reasoning trajectories. After each SFT epoch, we record the model's output trajectories and answers, then examine them to identify such cases. We then curate a special training subset $\mathcal{D}_{\text{reflect}}$ of all identified cases. For each case in $\mathcal{D}_{\text{reflect}}$, we pair $(R_{\text{early}}, R_{\text{late}})$, reconcile and consolidate the trajectories, yielding a ground-truth reasoning trajectory. We further fine-tune **Ophiuchus** on $\mathcal{D}_{\text{reflect}}$ using a training objective analogous to the cold-start SFT stage.

**Agentic Tool Reinforcement Learning.** We further optimize **Ophiuchus** through RL with carefully designed rewards, enabling the agent to adaptively discover effective tool-use strategies and achieve compositional tool use, thereby moving beyond the constraints of mimicking the static trajectories observed during SFT. ATRL process comprises three key components:

**(1) Rollout Formulation.** ATRL extends the traditional RL by introducing tool-call tokens and observation tokens. The observation tokens come from external function calls rather than the model itself. According to the formulation in Section 3.1, given a user query $Q$ and an image $I$, the reasoning trajectory up to step $k$ is denoted as: $R_k = (r_1, t_1, o_1), (r_2, t_2, o_2), ..., (r_k, t_k, o_k)$. At each step $k + 1$, the model must generate the next reasoning

thoughts $r_{k+1}$ and select a tool $t_{k+1}$ to formulate a parameterized invocation of the tool to make progress toward solving $Q$. The model's policy is defined as: $(r_{k+1}, t_{k+1}) \sim \pi_\theta(\cdot \mid I, Q, R_k)$. This rollout process continues to interleave until either an answer is generated or the maximum number of tool calls is reached. To prevent inefficient or circular behavior, we enable early termination: if a tool invocation duplicates a previously executed one, the rollout stops. We instruct the model to mark its thoughts, tool calls, and final answers in the output using the special tokens <think>, <tool_call>, and <answer>. When the model output includes <tool_call>, we automatically parse the tool calls into individual invocations using the model-predicted parameters. The outputs from executions are then inserted into the <obs> field and appended to the ongoing trajectory. All observation tokens are considered as a whole, which does not contribute to the loss computation. The system and user prompts are provided in Appendix F.3. Note that <tool_call> and <answer> are not mutually exclusive; they can co-occur within a single output.

**(2) Reward Design.** Unlike prior works (Guo et al., 2025; Lai et al., 2025b; Su et al., 2025b) that rely on overly simplified, answer-based rewards, we introduce a novel rule-based reward, decomposed into fine-grained signals that deliver dense feedback across various tool-use tasks, steering the model toward effective reasoning. Formally, the final reward comprises three components: reasoning format adherence, final answer quality, and strategic tool usage.

• The format reward $\mathcal{S}_{format}$ evaluates the structural validity of $R$ by verifying that the model's output includes all required special tokens in the prescribed order.

• The final-answer reward $\mathcal{S}_{ans}$ encompasses multiple task types. For close-ended questions, we simply check the exact match between the predicted and GT answers. For segmentation tasks, we compute the IoU between the predicted masks and GT masks and assign piecewise rewards based on predefined IoU thresholds.

• The strategic tool-use reward $\mathcal{S}_{tool}$ is defined as a conditional bonus, granted only when the model both produces a correct answer and invokes at least one external perception tool during its trajectory. This design encourages the model to employ tools meaningfully-when they directly contribute to successful task completion-rather than using them arbitrarily or redundantly.

The final reward $\mathcal{S}$ is derived as: $\mathcal{S} = \mathcal{S}_{ans} + \mathcal{S}_{format} + \mathcal{S}_{tool}$. Formal equations for each reward signal are provided in Appendix C. This diverse and fine-grained reward design better reflects the complexity of real-world tool use, guiding the model to generate outputs that are both syntactically valid and semantically faithful.

**(3) Optimization.** Based on the rollout formulation and rewards defined above, we optimize the policy using GRPO (Guo et al., 2025) without the KL penalty term (Hu et al., 2025) on dataset $\mathcal{D}_{rl}$:

$$\mathcal{L}_{\mathrm{RL}} = \mathbb{E}_{\substack{(I,Q,A)\sim\mathcal{D}_{\mathrm{rl}} \\ \{R_i\}_{i=1}^G \sim \pi_{\theta_{\mathrm{old}}}(\cdot|I,Q)}} \left( -\frac{1}{G}\sum_{i=1}^G \frac{1}{T_i}\sum_{n=1}^{N_i} \right.$$
$$\left. \min\left(\pi_{\theta_{i,n}}\mathcal{A}_i,\ \mathrm{clip}(\pi_{\theta_{i,n}}, 1-\epsilon,\, 1+\epsilon)\,\mathcal{A}_i\right) \right), \tag{2}$$

Here, $G$ is the number of rollout reasoning paths; $R_i = \{(r_{i,n}, t_{i,n}, o_{i,n})\}_{n=1}^{N_i}$ denotes the $i$-th reasoning path; $T_i$ is the total length of $R_i$ excluding tool outputs; $\mathcal{S}_i$ is the reward of $R_i$; and $\pi_\theta$ and $\pi_{\theta_{\mathrm{old}}}$ represent the current and old policy distributions, respectively. The normalized advantage $\mathcal{A}_i$ reflects the relative quality of each reasoning path within the rollout group, enabling the model to distinguish between learnable and poor reasoning trajectories. Through ATRL training, the agent learns to utilize tools for key visual cues inspection, mimic expert-like efficient reasoning behaviors.

### 3.3. Dataset Curation

**Data Collection and QA Generation.** Our data collection follows three core principles: (1) diverse tasks and imaging distributions; (2) scenarios where tool use yields measurable accuracy gains; and (3) comprehensive fine-grained annotations. Consequently, we leverage datasets from BiomedParseData (Zhao et al., 2024), which comprises 3.4 million triples of image, segmentation mask, and semantic label, encompass 82 major biomedical object types across 9 imaging modalities. We also include the Malenia dataset (Jiang et al., 2024), which provides 1,514 image-mask-report triplets with fine-grained descriptions of disease attributes across 12 lesion categories. To ensure that the tool invocations are genuinely necessary for resolving the vision-language queries, we prompt Gemini-2.5-pro (Comanici et al., 2025) to generate a QA pair conditioned on the image, mask, and the mask's descriptions, yielding a fine-grained question that necessitates localizing the specified mask region. We further eliminate data that cannot be properly verified, such as questions with incorrect answers. The prompts for VQA generation and verification are provided in Appendix F.1.

**Reasoning Trajectory Generation and Data Selection.** We leverage GPT-5 (OpenAI, 2025) to generate reasoning paths following our reasoning paradigm described in Section 3.1. During generation, whenever GPT-5 issues a tool call, we provide the corresponding GT outputs for that invocation. The generated reasoning paths are subsequently filtered based on format checking and answer correctness. Our data selection strategy encompasses two key steps: (1) We select instances where GPT-5 produces incorrect answers in single-turn interactions but achieves correct results when using tool invocation for fine-grained region inspection, highlighting scenarios where tool use is most beneficial. (2) We also include a small subset of questions that

*Table 1.* **Performance comparison on medical VQA benchmarks**. Gray-shaded rows denote large models. **Bold** and underlined indicate the best and second-best results, respectively. *Improvement* in the last row denotes the absolute gain of **Ophiuchus** over Qwen2.5-VL-7B (w/o tool use). Avg. is the mean over seven *Out-of-Domain* zero-shot benchmarks. For fairness, since **Med-R1-2B** is trained on part of the OmniMedVQA test set, its Avg is computed over the remaining six benchmarks (excluding OmniMed).

| Methods | Tool | In-domain $\mathcal{D}_{test}$-VQA | PathVQA | SLAKE | VQA-RAD | OmniMed | MMMU(H&M) | MedXpertQA | In-House-VQA | Avg. |
|---|---|---|---|---|---|---|---|---|---|---|
| *Close-Source SOTA* | | | | | | | | | | |
| GPT-4.1 | ✗ | 36.4 | 58.3 | 71.6 | 65.2 | 75.6 | 73.6 | 40.8 | 26.1 | 58.7 |
| GPT-5 | ✗ | 37.3 | 60.0 | 73.2 | 64.5 | 75.4 | 70.7 | 40.4 | 28.5 | 59.0 |
| OpenAI-o3 | ✓ | 39.8 | 67.5 | 75.3 | 66.0 | 73.7 | 74.5 | 44.1 | 30.7 | 61.7 |
| Gemini 2.5 Pro | ✓ | 40.2 | 67.1 | 72.7 | 63.8 | 76.9 | 72.8 | 46.6 | 30.3 | 61.5 |
| *Open-Source SOTA* | | | | | | | | | | |
| InternVL3-8B | ✗ | 34.9 | 53.2 | 70.4 | 65.6 | 72.2 | 62.3 | 23.8 | 20.4 | 52.6 |
| LLaVA-Next-13B | ✗ | 21.4 | 39.8 | 57.1 | 54.8 | 58.0 | 40.1 | 19.6 | 18.0 | 41.1 |
| Qwen2.5-VL-32B | ✗ | 36.5 | 47.7 | 70.1 | 71.7 | 69.5 | 60.1 | 26.8 | 25.5 | 53.1 |
| *MLLMs can "Think with Images"* | | | | | | | | | | |
| DeepEyes-7B | ✓ | 37.1 | 52.9 | 68.2 | 65.9 | 64.8 | 57.8 | 23.6 | 20.9 | 50.6 |
| Mini-o3-7B-v1 | ✓ | 37.7 | 53.4 | 67.8 | 65.7 | 65.1 | 57.4 | 24.3 | 21.5 | 50.7 |
| PixelReasoner-RL-v1-7B | ✓ | 37.4 | 52.6 | 67.3 | 66.0 | 64.9 | 58.0 | 23.5 | 21.2 | 50.5 |
| *Medical MLLMs* | | | | | | | | | | |
| LLaVA-Med-7B | ✗ | 25.3 | 44.6 | 47.7 | 52.5 | 52.1 | 38.8 | 19.9 | 19.3 | 39.3 |
| HuatuoGPT-Vision-34B | ✗ | 34.8 | 50.7 | 68.3 | 61.7 | 70.6 | 60.1 | 23.6 | 20.8 | 50.8 |
| *Medical MLLMs with CoT Reasoning* | | | | | | | | | | |
| Med-R1-2B* | ✗ | 17.5 | 19.2 | 52.1 | 36.5 | – | 44.7 | 22.9 | 17.1 | 32.1 |
| Lingshu-7B | ✗ | 38.2 | 68.4 | 77.8 | 66.4 | 74.9 | 57.8 | 25.2 | 23.8 | 56.3 |
| Chiron-o1-8B | ✗ | 38.6 | 68.8 | 77.4 | 72.5 | 76.2 | 55.4 | 24.3 | 25.9 | 57.2 |
| *Multimodal Medical Agents* | | | | | | | | | | |
| MMedAgent-7B | ✓ | 37.6 | 59.4 | 68.7 | 64.0 | 58.2 | 44.1 | 22.3 | 20.1 | 48.1 |
| MMedAgent-RL-7B | ✗ | 36.0 | 58.5 | 67.9 | 66.1 | 60.3 | 58.9 | 22.6 | 21.4 | 50.8 |
| VILA-M3-40B | ✓ | 39.5 | 66.4 | 71.4 | 65.7 | 55.1 | 56.6 | 23.0 | 25.3 | 51.9 |
| Qwen2.5-VL-7B | ✗ | 33.6 | 50.4 | 66.5 | 63.4 | 64.7 | 56.7 | 23.5 | 20.2 | 49.3 |
| **Ophiuchus-7B (ours)** | ✓ | **59.4**$^{(+25.8)}$ | **74.3**$^{(+23.9)}$ | **83.9**$^{(+17.4)}$ | **73.6**$^{(+10.2)}$ | **78.6**$^{(+13.9)}$ | **76.0**$^{(+19.3)}$ | **39.3**$^{(+15.8)}$ | **50.4**$^{(+30.2)}$ | **68.0**$^{(+18.7)}$ |

GPT-5 answers correctly without tools, aiming at teaching the model to adaptively employ tools only when necessary. Through systematic curation, we obtain 64k VQA samples spanning multiple task types and modalities. To ensure reliability, these samples are further cross-validated against human annotations. We split the data into 30k for $\mathcal{D}_{cold}$, 30k for $\mathcal{D}_{rl}$, and 4k for in-domain testing $\mathcal{D}_{test}$. Further details are provided in Appendix A. The prompts for reasoning trajectory generation and verification are provided in Appendix F.1.

## 4. Experiments

### 4.1. Experiment Settings

**Evaluation Benchmark and Metrics.** Beyond our in-domain-4k testing set, we also evaluate on several representative public benchmarks: (1) general medical VQA datasets—PathVQA (He et al., 2020), SLAKE (Liu et al., 2021), VQA-RAD (Lau et al., 2018), and OmniMed-VQA (Hu et al., 2024); (2) more challenging medical-reasoning benchmarks-MMMU(H&M) (Yue et al., 2024) and MedXpertQA (Zuo et al., 2025); and (3) a held-out in-house dataset collected from three medical centers, comprising 1k CT, MRI, and pathology images across 7 cancer types for out-of-domain testing. Benchmark details are provided in Appendix A. For evaluation metrics, we use answer accuracy for multiple-choice VQA benchmarks. For sub-

sets in $\mathcal{D}_{test}$-4k and in-house-1k that require segmentation results as answers, we use the DSC score.

**Implementation details.** We implement **Ophiuchus** based on Qwen2.5-VL-7B (Bai et al., 2025). The training is conducted on a cluster of 16 NVIDIA A100 GPUs. For the cold-start SFT stage, we optimize the model with a learning rate of $1 \times 10^{-5}$ for 10 epochs. The total batch size is 256. We use the same settings as in the cold-start SFT and conduct self-reflection fine-tuning for 2 epochs. The subsequent RL optimization is implemented using the VERL (Sheng et al., 2025) framework, where we set the training batch size to 256 and generate 8 candidate reasoning paths per question, up to a maximum of 6 times of tool calling. We use a constant learning rate of $1 \times 10^{-6}$ and set the maximum context length to 32K tokens. RL training runs for 12 epochs. During inference, we expose external model tools (*e.g.*, SAM2 and BiomedParse) as APIs using FastAPI (Ramírez & contributors, 2018–2025) for tool invocation acceleration.

### 4.2. Main Results

**Medical VQA Performance.** We conduct a comprehensive evaluation across eight benchmarks; results are summarized in Table 1. (1) Against general-purpose SOTA MLLMs-both closed-source (GPT-series (OpenAI, 2025), Gemini-2.5-Pro (Comanici et al., 2025)) and open-source (Qwen-series (Bai et al., 2025), LLaVA-Next-13B (Liu et al., 2024),

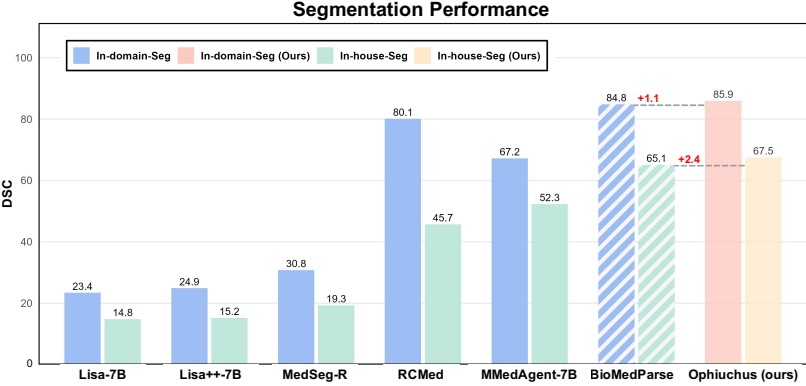

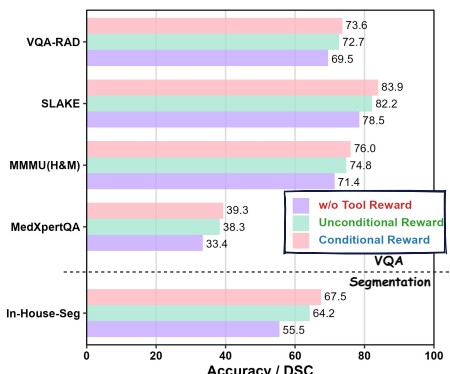

*Figure 3.* **Performance comparison on medical Segmentation benchmarks.** Colors indicate datasets (In-domain-Seg / In-house-Seg). We also provide performance of task-specific SOTA BiomedParse for reference.

*Figure 4.* **Ablation on Reward Design.** Results are shown separately for VQA and Segmentation.

*Table 2.* **Impact of Training Stages.** Checkmarks (✓) indicating which stages are used.

| Method | Training Stages | | | Benchmarks | | | | | |
|---|---|---|---|---|---|---|---|---|---|
| | SFT | Self-Reflection | ATRL | VQA-RAD | SLAKE | MMMU(H&M) | MedXpertQA | In-House-Seg | Avg. |
| *prompt-driven* $\mathcal{M}_{base}$ | | | | 63.7 | 67.6 | 58.5 | 23.7 | 24.9 | 47.7 |
| $\mathcal{M}_{cold}$ | ✓ | | | $67.6^{(+3.9)}$ | $76.7^{(+9.1)}$ | $66.2^{(+7.7)}$ | $30.3^{(+6.6)}$ | $52.6^{(+27.7)}$ | $58.7^{(+11.0)}$ |
| $\mathcal{M}_{cold-12}$ | ✓ | | | $67.6^{(+3.9)}$ | $76.8^{(+9.2)}$ | $66.3^{(+7.8)}$ | $30.3^{(+6.6)}$ | $52.5^{(+27.6)}$ | $58.8^{(+11.1)}$ |
| $\mathcal{M}_{cold+reflect}$ | ✓ | ✓ | | $68.9^{(+5.2)}$ | $78.0^{(+10.4)}$ | $70.9^{(+12.4)}$ | $32.0^{(+8.3)}$ | $54.2^{(+29.3)}$ | $60.8^{(+13.1)}$ |
| $\mathcal{M}_{rl}$ | | | ✓ | $71.3^{(+7.6)}$ | $79.5^{(+11.9)}$ | $71.5^{(+13.0)}$ | $34.5^{(+10.8)}$ | $57.3^{(+32.4)}$ | $62.8^{(+15.1)}$ |
| $\mathcal{M}_{cold+rl}$ | ✓ | | ✓ | $\underline{72.5}^{(+8.8)}$ | $\underline{81.3}^{(+13.7)}$ | $\underline{73.7}^{(+15.2)}$ | $\underline{37.1}^{(+13.4)}$ | $\underline{63.9}^{(+39.0)}$ | $\underline{65.7}^{(+18.0)}$ |
| **Ophiuchus** | ✓ | ✓ | ✓ | $\mathbf{73.6}^{(+9.9)}$ | $\mathbf{83.9}^{(+16.3)}$ | $\mathbf{76.0}^{(+17.5)}$ | $\mathbf{39.3}^{(+15.6)}$ | $\mathbf{67.5}^{(+42.6)}$ | $\mathbf{68.1}^{(+20.4)}$ |

InternVL3-8B (Zhu et al., 2025a))-**Ophiuchus** achieves substantially better performance. (2) Compared with SOTA MLLMs that also can "think with images" via zoom-in operations (DeepEyes-7B (Zheng et al., 2025), Mini-o3-7B-v1 (Lai et al., 2025a), and PixelReasoner (Su et al., 2025a)), our method achieves consistently superior performance across all datasets, underscoring the effectiveness of incorporating external segmentation models as tools. (3) Compared with SOTA medical-specific MLLMs (LLaVA-Med-7B (Li et al., 2023) and HuatuoGPT-Vision-34B (Chen et al., 2024a)) and medical reasoning MLLMs (*e.g.*, Med-R1-2B (Lai et al., 2025b), Chiron-o1-8B (Sun et al., 2025a), and Lingshu-7B (Xu et al., 2025)), **Ophiuchus** achieves at least a 10.8% average improvement, validating that tool-augmented reasoning enhances MLLMs' ability to capture fine-grained information and leverage key visual evidence for medical question answering. (4) Relative to SOTA medical MLLM agents that also can invoke tools (MMedAgent-7B (Li et al., 2024), VILA-M3-40B (Nath et al., 2025)) or rely on multi-agent collaboration (MMedAgent-RL-7B (Xia et al., 2025)), **Ophiuchus** outperforms competing agents by a substantial margin, further validating the effectiveness of our tool-augmented vision-language reasoning. Rather than merely calling tools to complete a task, we integrate tool use into step-by-step reasoning, enabling the model to compose multiple tools and reflect on tool outputs for better

visual understanding.

**Segmentation Performance.** We further evaluate segmentation to validate **Ophiuchus** on fine-grained regional features. We report results on In-domain-Seg and In-house-Seg, two subsets where VQA items explicitly require pixel-level masks. Figure 3 compares **Ophiuchus** with general-purpose segmentation MLLMs Lisa (Lai et al., 2024) and Lisa++ (Yang et al., 2023), medical reasoning-based segmentation MLLMs MedSeg-R (Huang et al., 2025) and RCMed (Wang et al., 2025a), and the tool-using medical agent MMedAgent-7B (Li et al., 2024). **Ophiuchus** achieves the best performance on In-domain-Seg. To mitigate data-leakage concerns from models trained on large public datasets such as SAM2, we also evaluate on the fully held-out In-house-Seg set, where **Ophiuchus** remains ahead and improves over prior work by at least 15.2%. We observe that tool-enabled agents generally outperform MLLMs trained to segment without tools, and that **Ophiuchus** gains beyond tool strength by selecting among similar tools and reflecting on their outputs, switching to alternatives when needed. Additional analyses on tool quantity and type are provided in Table 6, with further segmentation-tool variants in Appendix D.2.

We also present a qualitative analysis of the diverse reasoning patterns that emerge in **Ophiuchus**, illustrating how the

*Table 3.* **Comparison with multi-agent systems.**

| Method | $\mathcal{D}_{test}$-VQA | SLAKE |
|---|---|---|
| Multi-Agent Pipeline (BioMedParse + QwenVL-2.5-7B) | 37.2 | 68.9 |
| GT region crop + QwenVL-2.5-7B | 39.9 | 70.8 |
| GT region crop + LLaVA-Med-7B | 33.8 | 53.4 |
| **Ophiuchus (ours)** | **59.4** | **83.9** |

*Table 4.* **Comparison with static crop-based baselines.**

| Benchmark | Qwen2.5-VL-7B | + self-crop bbox | + GT bbox | Ophiuchus |
|---|---|---|---|---|
| PathVQA | 50.4 | 52.8 | 58.7 | **74.3** |
| SLAKE | 66.5 | 67.6 | 70.8 | **83.9** |
| VQA-RAD | 63.4 | 63.7 | 67.5 | **73.6** |
| OmniMed | 64.7 | 65.1 | 67.2 | **78.6** |
| MMMU | 56.7 | 58.5 | 60.1 | **76.0** |
| MedXpertQA | 23.5 | 23.7 | 28.2 | **39.3** |
| In-House | 20.2 | 20.3 | 31.5 | **50.4** |
| Avg. | 49.3 | 50.2 | 54.9 | **68.0** |

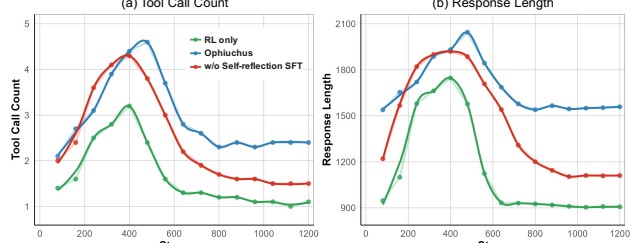

*Figure 5.* **Training Dynamics.**

model integrates visual tools into its reasoning-analogous to expert visual cognition-and adapts their use to task demands. Cases are provided in the Appendix E.

### 4.3. Ablation Studies

**Effectiveness of training strategies.** To assess the impact of our training framework, we compare **Ophiuchus** with the following Qwen2.5-VL-7B-based variants: (1) a prompt-driven tool-invocation baseline; (2) $\mathcal{M}_{cold}$, trained only with cold-start SFT; (3) $\mathcal{M}_{cold+reflect}$, trained with cold-start SFT followed by self-reflection fine-tuning; (4) $\mathcal{M}_{rl}$, trained only with ATRL; and (5) $\mathcal{M}_{cold+rl}$. The results in Table 2 validate the effectiveness of our three-stage training strategy. Prompt-only method for tool invocation is neither adaptable nor robust and yields minimal gains, whereas progressively adding our proposed training stages produces substantial improvements over the base model, underscoring the necessity of each stage. To further demonstrate the effectiveness of self-reflection fine-tuning, we compare $\mathcal{M}_{cold+reflect}$ (trained for $10 + 2$ epochs) with a baseline model $\mathcal{M}_{cold-12}$, which undergoes 12 epochs of standard SFT on the original $\mathcal{D}_{cold}$ dataset. The results confirm that the subsequent improvements primarily stem from fine-tuning on the self-reflection data rather than merely increasing the number of SFT epochs. The RL stage yields the largest gains, indicating that reward-driven exploration and feedback are essential for learning context-aware tool-use policies and strategic tool orchestration. Moreover, combining RL with cold-start SFT and self-reflection maximizes these benefits, supporting our three-stage training framework for tool-augmented reasoning.

To visualize behavioral evolution, we track the mean number of tool calls and response length during training in Figure 5. **Ophiuchus** exhibits an expand–compress trend: both increase early as the model explores tools and hypotheses, then decrease as RL consolidates experience, invokes only necessary tools, and terminates once evidence is sufficient. Without self-reflection, early expansion is weaker, while the RL-only variant underuses tools throughout. Overall, our three-stage recipe promotes broad exploration and then distills it into efficient, context-aware tool orchestration.

**When does Ophiuchus invoke tools?** We further analyze whether the learned policy invokes tools selectively rather than blindly. For each benchmark, we first identify cases

that the base Qwen2.5-VL-7B fails to answer without tools, where tool invocation is considered necessary. As shown in Table 5, **Ophiuchus** invokes tools and answers correctly on 93.7% of such cases on average, with all benchmarks above 90%. Conversely, for cases that the base model can already answer correctly without tools, **Ophiuchus** rarely makes unnecessary tool calls, with an average rate of only 3.3%. These results show that **Ophiuchus** learns a selective tool-use policy: it invokes tools when fine-grained visual evidence is needed, while avoiding redundant calls when the global image is sufficient.

**Importance of the strategic tool-use reward.** Our strategic tool-use reward includes a conditional tool-invocation bonus granted only when the model answers correctly and employs tools. This mechanism incentivizes effective, purposeful tool use. We ablate the reward by (i) removing it entirely (w/o tool reward) and (ii) making it unconditional on accuracy. Results in Figure 4 show that omitting the tool reward causes a substantial performance drop, underscoring its importance. Moreover, the conditional reward yields the highest accuracy, outperforming the other settings. These findings indicate that rewarding tool use alone is insufficient; aligning rewards with successful outcomes is what truly drives intelligent and effective behavior in **Ophiuchus**.

**Is the gain merely from tool access or local crops?** We further examine whether the improvement simply comes from access to external tools or enlarged local regions. First, we compare **Ophiuchus** with workflow-style multi-agent variants, including BioMedParse + Qwen2.5-VL-7B, which pre-segments and enlarges target regions before analysis, and GT-crop baselines where Qwen2.5-VL-7B and LLaVA-Med-7B receive ground-truth box crops. As shown in Ta-

*Table 5.* **Analysis of necessary and unnecessary tool-calling behaviors.**

| Behavior | $\mathcal{D}_{test}$-VQA | PathVQA | SLAKE | VQA-RAD | OmniMed | MMMU | MedXpertQA | In-House | Avg. |
|---|---|---|---|---|---|---|---|---|---|
| Necessary tool-calling | 94.7 | 92.5 | 97.1 | 96.7 | 94.6 | 93.0 | 90.8 | 90.4 | 93.7 |
| Unnecessary tool-calling | 4.8 | 1.4 | 4.2 | 0.6 | 3.5 | 3.7 | 3.9 | 4.6 | 3.3 |

*Table 6.* **Ablation on tool quantity and type.** Tools are grouped into seen and unseen. Checkmarks indicate the tools used in each setting.

| Method | Tools # | Tools | | | | | Benchmarks | | | | | |
|---|---|---|---|---|---|---|---|---|---|---|---|---|
| | | Seen | | | Unseen | | VQA tasks | | | | Segmentation | Avg. |
| | | Zoom-in | SAM2 | BioMed Parse | MedSAM | | VQA-RAD | SLAKE | MMMU (H&M) | MedXpertQA | In-House-Seg | |
| DeepEyes-7B | 1 | ✓ | | | | | 65.9 | 68.2 | 57.8 | 23.6 | – | 53.9 |
| Mini-o3-7B-v1 | 1 | ✓ | | | | | 65.7 | 67.8 | 57.4 | 24.3 | – | 53.8 |
| PixelReasoner-RL-v1-7B | 1 | ✓ | | | | | 66.0 | 67.3 | 58.0 | 23.5 | – | 53.7 |
| MMedAgent-MedSAM | 1 | | | | ✓ | | 64.0 | 68.7 | 44.1 | 22.3 | 52.3 | 50.3 |
| Ophiuchus | 1 | ✓ | | | | | 68.0 | 72.5 | 60.5 | 29.3 | – | 57.6 |
| | 1 | | | | ✓ | | 69.5 | 78.2 | 71.4 | 31.0 | 63.9 | 62.8 |
| | 1 | | ✓ | | | | 70.5 | 78.4 | 71.6 | 31.2 | 64.1 | 63.2 |
| | 1 | | | ✓ | | | 71.6 | 79.8 | 73.5 | 33.7 | 64.5 | 64.6 |
| | 2 | | ✓ | ✓ | | | 71.9 | 80.4 | 73.8 | 34.4 | 67.0 | 65.5 |
| | 2 | ✓ | ✓ | | | | 72.4 | 81.9 | 74.1 | 37.0 | 64.8 | 66.0 |
| | 2 | ✓ | | ✓ | | | 72.8 | 82.7 | 75.0 | 38.4 | 65.0 | 66.8 |
| **Ophiuchus (all tools)** | **3** | ✓ | ✓ | ✓ | | | **73.6** | **83.9** | **76.0** | **39.3** | **67.5** | **68.1** |

ble 3, **Ophiuchus** substantially outperforms these static pipelines, indicating that pre-defined segmentation or cropping workflows are insufficient for robust tool-augmented reasoning. To further isolate the effect of localized visual evidence, we construct two additional crop-based baselines using Qwen2.5-VL-7B: a self-crop baseline that first predicts the target bounding box and then answers from the cropped image, and a GT-crop baseline that receives ground-truth object crops. As shown in Table 4, self-cropping brings only marginal gains over the base model, and even GT crops remain substantially below **Ophiuchus**. These results suggest that the key capability is not merely access to tools or enlarged local regions, but the ability to iteratively decide what evidence is needed, invoke tools with context, and integrate tool-returned observations into multimodal reasoning.

### 4.4. Discussion: Tool Quality *vs.* Tool Quantity

We emphasize that tool quality and coverage are more important than raw tool count. Our objective is to elicit pixel-level reasoning that improves an MLLM's understanding of fine-grained visual cues in medical images, rather than to accumulate as many tools as possible. In particular, SAM 2 and BioMedParse are strong foundation models that each support a broad spectrum of segmentation tasks across image modalities and anatomical structures. Combined with a zoom-in operator, these tools provide sufficient functionality for fine-grained visual analysis. Therefore, selecting these three tools reflects an intentional and careful design choice, rather than a limitation in the number of integrated tools. Moreover, Table 6 and Table 9 show that Ophiuchus can leverage segmentation tools in a zero-shot manner, indicating that the current tool set is extensible and should not be viewed as an upper bound. The key lies in tool complementarity and the depth and breadth of each tool's capabilities, rather than simply counting how many tools are included.

## 5. Conclusion

In this paper, we propose **Ophiuchus**, a new medical MLLM agent with the ability of interleaved multimodal reasoning through invoking external tools for fine-grained visual perception and emulation of expert-like diagnostic behavior in medical image analysis. Unlike simple tool-calling, **Ophiuchus** offers several key advantages, including advanced tool orchestration, enhanced generalization, and multimodal CoT capabilities. Through careful empirical validation, we demonstrate that our three-stage training framework successfully cultivates sophisticated reasoning patterns. We hope **Ophiuchus** can serve as a foundation for advanced tool-augmented visual reasoning, helping the community develop medical AI agents that can genuinely "think with images" using tools.

## Acknowledgement

This work was supported by the Zhejiang Provincial Natural Science Foundation of China under the Exploratory Youth Program (Grant No. LQ23H160038) awarded to S.Z. We thank the LeapQuest AI team at Shanghai Innovation Institute and our collaborators for their valuable support and discussions. We also sincerely appreciate the anonymous reviewers and the Area Chair for their constructive feedback,

which helped improve this work.

## Impact Statement

This work does not involve experiments with human subjects, animal testing, or personally identifiable information. All training datasets used in this study are derived from publicly available sources. Regarding the in-house test dataset, this retrospective study was approved by the ethics committee of our collaborating hospital, and the requirement for informed consent was waived. All personally identifiable information and private details were removed. The datasets, methodologies, and results presented here do not pose foreseeable risks of harm. This paper presents work whose goal is to advance the field of Machine Learning.

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

# A. Dataset Details

## A.1. Details of our Curated Datasets

Our data collection follows three core principles: (1) Diverse Tasks and Imaging Distributions. We incorporate varied data to strengthen model generalization. (2) Tool Effectiveness. We prioritize scenarios in which tool use yields measurable accuracy gains. (3) Comprehensive Fine-Grained Annotations. We select datasets that include fine-grained region masks and corresponding descriptions, providing the supervision needed to improve the model's visual reasoning. Consequently, we leverage datasets from BiomedParseData (Zhao et al., 2024), which comprises 3.4 million triples of image, segmentation mask, and semantic label, encompass 82 major biomedical object types across 9 imaging modalities. We also include the Malenia dataset (Jiang et al., 2024), which provides 1,514 image-mask-report triplets across 12 lesion categories with fine-grained descriptions of disease attributes including location, lesion shape, density, density variations, and surface characteristics. These two datasets provide rich fine-grained region annotations and detailed textual descriptions. This abundance of information enables us to construct precise VQA samples and reasoning trajectories. Based on these two datasets, we synthesize queries that specifically require the localization of fine-grained visual cues. Concretely, we prompt Gemini-2.5-pro (Comanici et al., 2025) to generate a QA pair conditioned on the image, mask, and the mask's descriptions, yielding a fine-grained question that necessitates localizing the specified mask region. We further eliminate data that cannot be properly verified, such as questions with incorrect answers. The prompts for VQA generation and verification are provided in Figure 17 and Figure 18. As described in Section 3.3, we then leverage GPT-5 (OpenAI, 2025) to generate reasoning paths following our proposed tool-augmented reasoning paradigm. The generated paths are subsequently filtered based on format checking and answer correctness. The prompt template for constructing reliable agentic reasoning trajectories and the prompt template for trajectory verification are provide in Figure 19 and Figure 20.

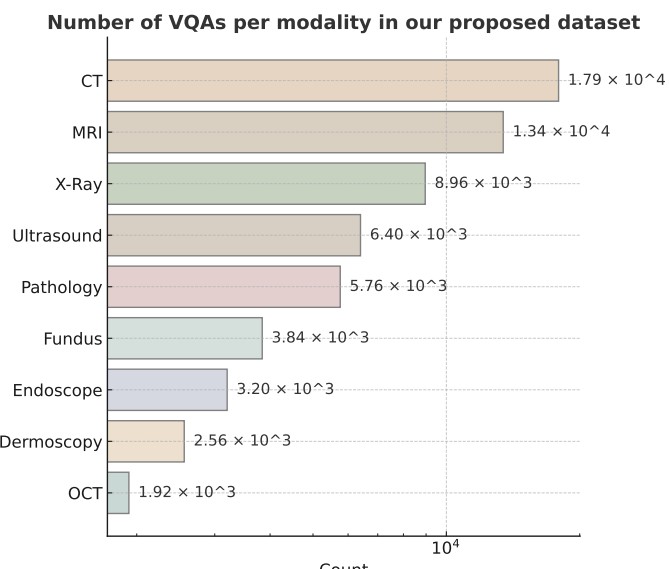

*Figure 6.* **Distribution of modalities in our proposed dataset.**

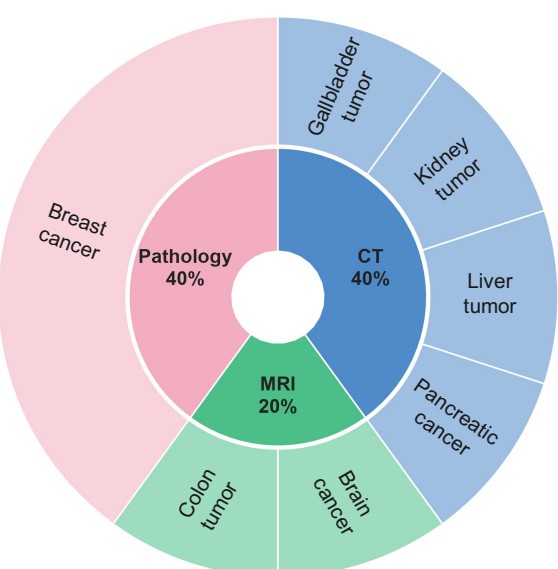

*Figure 7.* **Distribution of modalities and disease types in the in-house dataset.**

Our data selection strategy encompasses two key steps:(1) We select instances where GPT-5 produces incorrect answers in single-turn interactions but achieves correct results when using tool invocation for fine-grained region inspection, highlighting scenarios where tool use is most beneficial. (2) We also include a small subset of questions that GPT-5 answers correctly without tools, aiming at teaching the model when to rely on its internal capabilities instead of always invoking external tools. Through systematic curation, we obtain 64k VQA samples spanning multiple task types and modalities. To ensure reliability, these samples are further cross-validated against human annotations. Our dataset encompasses a highly diverse set of question types, including: multi-organ and multi-disease recognition and localization; determination of the presence of abnormal lesions; classification and counting of lesion categories; assessment of lesion characteristics, morphology, and intensity variations; as well as organ and lesion segmentation—challenging tasks that more closely reflect real-world clinical diagnosis. Figure 6 presents a bar plot of the number of VQA items for each modality in our dataset. The dataset comprises nine medical imaging modalities: histopathology (9%), CT (28%), MRI (21%), ultrasound (10%), X-ray (14%), fundus

*Table 7.* Numbers of VQA samples and segmentation samples in testing sets.

| VQA tasks | | Segmentation | |
|---|---|---|---|
| $\mathcal{D}_{\text{test}}$-**VQA** | **In-House-VQA** | $\mathcal{D}_{\text{test}}$-**Seg** | **In-House-Seg** |
| 3140 | 653 | 860 | 347 |

photography (6%), dermoscopy (4%), endoscopy (5%), and optical coherence tomography (OCT; 3%). For each modality, we split the data in fixed proportions: 30k samples for $\mathcal{D}_{cold}$ used in cold-start supervised fine-tuning, 30k samples for $\mathcal{D}_{rl}$ used in reinforcement learning training, and 4k for in-domain testing $\mathcal{D}_{test}$. Within $\mathcal{D}_{cold}$ (used for cold-start supervised fine-tuning), we apply the self-reflection sampling strategy introduced in Section 3.2 to select cases exhibiting self-correction behaviors. This yields a dataset $\mathcal{D}_{reflect}$ for self-reflection fine-tuning comprising 3k VQA instances.

### A.2. Details of the In-House Testset

For testing, in addition to our in-domain 4k testset $\mathcal{D}_{\text{test}}$, we use a completely held-out in-house dataset comprising 1k CT, MRI, and histopathology images across 7 cancer types for out-of-domain evaluation. In this held-out set, CT images include 100 liver tumor cases, 100 gallbladder tumor cases, 100 pancreatic cancer cases, and 100 kidney tumor cases. MRI images include 100 colon tumor cases and 100 brain cancer cases. Histopathology images include 400 breast cancer cases. Figure 7 depicts the data distribution of the in-house dataset. For each case, human annotators construct a VQA pair along with a corresponding reasoning trajectory. To evaluate our method on both VQA and segmentation tasks, we explicitly require the model to output segmentation results for a subset of questions, rather than using segmentation solely as an intermediate step for reasoning. This further partitions the test set into VQA and segmentation subsets. Table 7 reports the exact counts of VQA and segmentation questions in both the in-domain $\mathcal{D}_{\text{test}}$ and the in-house test set.

### A.3. Introduction of Public Benchmarks

We also conduct experiments on 6 representative datasets. PathVQA (He et al., 2020), SLAKE (Liu et al., 2021), and VQA-RAD (Lau et al., 2018) are widely used benchmarks in medical VQA research. OmniMedVQA (Hu et al., 2024) constructs multiple medical classification datasets into QA form, focusing on image classification tasks. For higher-level reasoning, MMMU-Med (Yue et al., 2024) and MedXpertQA (Zuo et al., 2025) offer more challenging QA scenarios. MMMU-Med is a medical subset extracted from the multimodal reasoning benchmark MMMU (Yue et al., 2024). MedXpertQA presents a more difficult setting, simulating real clinical licensing exams to assess whether models can perform medical reasoning and decision-making at a near-expert level.

### A.4. Filtering and Verification Pipeline for QA Pairs and Reasoning Traces

Regarding QA generation, we provide Gemini-2.5-Pro with the original image, the ground-truth mask, and the corresponding class label, and explicitly instruct it only to convert the existing mask and label into a QA format, without introducing any additional information. After QA generation, we perform a second layer of strict verification using GPT-5, which evaluates each QA item along the following five dimensions:

1. **Schema & Format**

   - JSON format must be valid and parseable.
   - Required keys `Question`, `Options`, and `Answer` must be present.
   - Options must contain four or five choices labeled strictly A–D (and optionally E).

2. **Independence & Wording**

   - The question must be self-contained and avoid referential expressions.
   - Wording must use professional, precise clinical terminology with no vague or colloquial phrasing.

3. **Grounding to the Target Region**

   - The question must be answerable from the image with focused attention on the segmented target region (guided by the bounding box and mask).

- High semantic correlation with the masked area is required; content must pertain to structures or findings within or immediately adjacent to the target region.

- No hallucinated information is allowed—only content derivable from the image, mask, and mask descriptions is permitted (no modality, demographic, or history hallucinations unless explicitly provided).

4. **Options Quality**

- Options must be mutually exclusive, non-overlapping, and clinically plausible.
- No duplicate or near-duplicate options.

5. **Answer Correctness**

- GPT-5 re-evaluates the image with the bounding box/mask and mask descriptions to determine the true correct answer.
- It verifies that the provided `Answer` corresponds to the correct option.

The full verification prompt used by GPT-5 for this VQA quality check is provided in Figure 18.

Regarding the generated reasoning traces, we also apply a rigorous verification procedure. Specifically, we use GPT-o3 to examine each trace according to the following evaluation rules:

- **Format:** The trace must interleave `<think>` → `<tool_call>` → `<obs>` steps (repeated as needed) and conclude with `Action: Answer` and an `<answer>` block.

- **Tool usage:** Only the allowed tools (`image_zoom_in_tool`, `SAM2`, `BioMedParse`) may be invoked, and all tool calls must contain valid JSON arguments.

- **Tool choice rationale:** Each `<tool_call>` must be explicitly justified by the immediately preceding `<think>` step (i.e., why this tool is selected and why these arguments are appropriate).

- **Observation integration:** Each `<obs>` must be meaningfully incorporated into the subsequent `<think>`; the model should not ignore or contradict the tool outputs.

- **Clinical correctness:** The reasoning must be consistent with the image/ROI/mask and the mask description, with no hallucinated findings or unsupported clinical claims.

- **Final answer:** The content of `<answer>` must exactly match the provided ground-truth answer.

All tool calls in the reasoning traces are also executed in the actual environment, and the resulting outputs are used to verify their consistency, for example, checking whether the mask covers the correct region. Any hallucinated content that conflicts with the underlying structured annotations is removed from the dataset. Through this ground-truth–guided trajectory construction and stepwise trajectory validation, we ensure that all retained data are of consistently high quality. The full verification prompt used by GPT-o3 for the reasoning trace quality check is provided in Figure 20.

After all the above checks, all QA data and reasoning trajectories are further cross-verified by human annotators.

## B. Cold-Start SFT Objective

We minimize the average negative log-likelihood over all reasoning and tool-call tokens:

$$\mathcal{L}_{\text{cold}} = \mathbb{E}_{(I,Q,A,R=\{(r_n,t_n,o_n)\}_{n=1}^{N}) \sim \mathcal{D}_{\text{cold}}} \left( -\frac{1}{T} \sum_{n=1}^{N} \log p(r_n, t_n \mid I, Q, R_{<n}) \right), \tag{3}$$

where $T = \sum_{n=1}^{N} (|r_n| + |t_n|)$ is the total number of tokens in reasoning steps and tool calls, and $A$ denotes the ground-truth answer.

## C. Formula Definition of Reward Signals

We adopt a diverse set of reward designs to provide rich and fine-grained feedback signals. The specific definitions of each reward are detailed as follows.

**The reasoning-format reward.** The reasoning-format reward $\mathcal{S}_{format}$ evaluates the structural validity of $R$ by verifying that the model's output includes all required special tokens in the prescribed order. Specifically, the model should enclose its chain-of-thought between `<think>` and `<think>` tags, place the tool-call choices and parameters between `<tool_call>` and `<tool_call>` tags, and place the final answer between `<answer>` and `<answer>` tags. Outputs that adhere to this structure receive a positive reward.

$$\mathcal{S}_{\text{format}} = \begin{cases} 1, & \text{if all required fields appear and are in the correct order,} \\ 0, & \text{otherwise.} \end{cases} \tag{4}$$

**The final-answer reward.** The final-answer reward $\mathcal{S}_{ans}$ encompasses multiple task types, thereby providing the agent with diverse feedback. For multiple-choice questions, we simply check the exact match between the predicted and answers:

$$S_{\text{ans}}(A, \hat{A}) = \mathbb{I}\left(A = \hat{A}\right), \text{for multiple-choice questions,} \tag{5}$$

$A$ denote the ground-truth answer and $\hat{A}$ is the predicted answer obtained by rule-based parsing of the model's final output. The indicator $\mathbb{I}$ is defined to be 1 if $A = \hat{A}$ and 0 otherwise.

For segmentation tasks, in contrast to earlier reward designs, we use SAM2 or BiomedParse as external reward providers. Given either a target location (bounding box) or a text prompt specifying the object category predicted by the MLLM, we query SAM2 or BiomedParse to obtain a segmentation mask. We then compute the intersection-over-union (IoU) between this mask and the ground-truth mask and assign piecewise rewards as follows:

$$\text{reward} = \begin{cases} 3, & \text{IoU} > 0.80, \\ 2, & 0.70 < \text{IoU} \le 0.80, \\ 1, & 0.50 < \text{IoU} \le 0.70, \\ 0, & \text{otherwise.} \end{cases} \tag{6}$$

This segmentation reward supplies strong positive feedback only when the predicted region closely matches the ground truth, while at lower IoU levels it provides guidance that encourages gradual improvement.

**The strategic tool-use reward.** The strategic tool-use reward $\mathcal{S}_{tool}$ is defined as a conditional bonus, granted only when the model both produces a correct answer and invokes at least one external perception tool during its trajectory. This design encourages the model to employ tools meaningfully-when they directly contribute to successful task completion-rather than using them arbitrarily or redundantly.

$$\mathcal{S}_{tool} = \mathbb{I}\left(\mathcal{S}_{ans} > 0\right) \cdot B_{\text{bonus}}, \tag{7}$$

where $\mathbb{I}\left(\mathcal{S}_{ans} > 0\right)$ is the indicator function that equals 1 only when $\mathcal{S}_{ans} > 0$, and $B_{\text{bonus}}$ represents the tool-invocation bonus, fixed at 2.

## D. Additional Evaluation Results

### D.1. The accuracy rate of tool invocation

Table 8 measures reliability by jointly checking format adherence and correct tool invocation. The prompt-driven baseline is unreliable, averaging 25.1% across VQA and segmentation. **Ophiuchus** is near ceiling on all settings with a 97.9% average, reflecting consistently precise tool-use behavior on both in-domain and in-house data. The average absolute gain is about 73%. These results show that the model reliably and proactively invokes tools to solve problems. This further corroborates the importance of our carefully designed multi-regime training for eliciting robust tool-calling capabilities—intelligent behaviors that mere prompting of a base model cannot achieve.

### D.2. Ablation on Tool Quantity and Type

This section studies how the quantity and type of tools affect performance. Results are shown in Table 6.

**Zoom-in-only OPHIUCHUS remain strong.** For a fair comparison with DeepEyes, Mini-o3, and PixelReasoner, we enable

*Table 8.* **Accuracy of Tool Use.** Values are percentages. Accuracy is computed by checking whether the model adheres to the required output format and correctly invokes the specified tools. **Avg.** is the arithmetic mean across the four columns.

| Method | VQA tasks | | Segmentation | | Avg. |
|---|---|---|---|---|---|
| | $\mathcal{D}_{test}$-VQA | In-House-VQA | $\mathcal{D}_{test}$-Seg | In-House-Seg | |
| prompt-driven $\mathcal{M}_{base}$ | 22.8 | 23.5 | 27.9 | 26.2 | **25.1** |
| **Ophiuchus** | 97.2 | 97.7 | 98.4 | 98.1 | **97.9** |

only the zoom-in tool in Ophiuchus. The zoom-in-only OPHIUCHUS reaches a 57.6 average, outperforming the zoom-in-only baselines that cluster near 53.8. It also leads on VQA-RAD with 68.0 compared with 65.7-66.0 for prior work, and maintains advantages on SLAKE and MMMU(H&M). This indicates that our training induces a deeply tool-integrated reasoning capability even when only local magnification is available.

**Training-free generalization to unseen tools.** When provided only with the unseen tool MedSAM, **Ophiuchus** achieves an average score of 62.8, which is substantially higher than the pipeline-style MMedAgent that also relies on MedSAM (50.3). The advantage holds consistently across VQA-RAD, SLAKE, MedXpertQA, and In-House-Seg. For instance, **Ophiuchus** reaches 69.5 compared with 64.0 on VQA-RAD and 63.9 compared with 52.3 on In-House-Seg. This demonstrates that **Ophiuchus** can recognize and exploit a new tool in a training-free manner, showing that its tool use is not restricted to the three tools introduced during development.

**Tool scaling and complementarity drive consistent gains.** Performance improves steadily as the model gains access to more tools, reflecting both a scaling effect and meaningful complementarity. Among single-tool settings, BiomedParse achieves the strongest average at 64.6, slightly ahead of SAM2 at 63.2, and both provide clear gains over zoom-in alone. When combined, the two segmentation tools reach an average of 65.5 and further raise In-House-Seg to 67.0, suggesting improved robustness through their complementary strength. Pairing zoom-in with segmentation tools also strengthens performance, with zoom-in plus SAM2 averaging 66.0 and zoom-in plus BiomedParse averaging 66.8. On MedXpertQA, the increase from 31.2 with SAM2 alone to 38.4 with zoom-in plus BiomedParse illustrates how local inspection and text-driven segmentation reinforce one another in complex reasoning. Equipping the agent with all three tools yields the best overall performance, averaging 68.1 and delivering improvements across both VQA and segmentation. These results demonstrate that broader tool access systematically scales performance, while diverse functionalities interact synergistically to expand the available evidence for reasoning.

**Adaptive decision-making across tools is essential.** Beyond the benefits of scale and diversity, the results also suggest that performance gains arise from the agent's ability to adaptively decide when and how to use each tool. The consistent improvements across different combinations indicate that the model does not rely on a single dominant utility, but instead learns to coordinate multiple perceptual pathways depending on task demands. This highlights that future progress will hinge not only on expanding tool libraries but also on strengthening adaptive orchestration strategies that allow the agent to dynamically align tools with the underlying reasoning process.

**D.3. Impact of Changing Segmentation Tools on the Agent's Performance.**

To investigate how sensitive Ophiuchus is to the inherent accuracy of its underlying segmentation tools (e.g., SAM2), we evaluate how sharply its performance changes when these tools are replaced with alternative ones. To this end, we conduct experiments in which SAM2—the default segmentation module in Ophiuchus—is replaced with SAM (Kirillov et al., 2023), MedSAM (Ma et al., 2024), SAM-Med2D (Ye et al., 2023), or MedSAM2 (Ma et al., 2025), respectively. Because these tools offer similar functionalities and support the same input parameter formats, Ophiuchus can use them in a zero-shot manner without requiring any additional training. The results are summarized in Table 9. Our comparisons show that replacing SAM 2 with different SAM-based segmentation tools results in only minor performance variations. This is due to Ophiuchus's strong visual feature understanding and grounding capabilities: as long as the MLLM provides an accurate bounding box as the visual prompt, these SAM-based tools can generally produce reliable segmentation outputs. This observation further underscores the robustness of Ophiuchus. Moreover, more capable segmentation tools naturally produce better segmentation performance. Ophiuchus possesses sufficient tool-understanding to directly leverage upgraded segmentation tools as long as their functional descriptions and parameter specifications are provided, thereby eliminating the need for any retraining.

*Table 9.* Ablation study of segmentation tool variants integrated into Ophiuchus. Default setting (SAM2) is highlighted.

| Method | VQA-RAD | SLAKE | MMMU (H&M) | MedXpertQA | In-House-Seg | Avg. |
|---|---|---|---|---|---|---|
| Ophiuchus + MedSAM (Ma et al., 2024) | 72.9 | 83.1 | 75.4 | 38.8 | 66.5 | 67.3 |
| Ophiuchus + MedSAM2 (Ma et al., 2025) | 73.6 | 83.8 | 75.9 | 39.5 | 67.3 | 68.0 |
| Ophiuchus + SAM-Med2D (Ye et al., 2023) | 73.3 | 83.6 | 75.7 | 38.9 | 67.1 | 67.7 |
| Ophiuchus + SAM (Kirillov et al., 2023) | 73.1 | 83.3 | 75.5 | 38.6 | 66.8 | 67.5 |
| **Ophiuchus + SAM2 (default)** | **73.6** | **83.9** | **76.0** | **39.3** | **67.5** | **68.1** |

*Table 10.* Segmentation performance of tools.

| Method | In-Domain-Seg | In-House-Seg |
|---|---|---|
| SAM 2 | 83.9 | 65.8 |
| BioMedParse | 84.8 | 65.1 |
| SAM 2 + BioMedParse (ensemble) | 85.3 | 66.2 |
| Ophiuchus-7B (ours) | 85.9 | 67.5 |

## D.4. Do the Tool Performances Constitute an Upper Limit for Ophiuchus?

Table 10 presents a comparison between Ophiuchus and the standalone performance of the segmentation tools it relies on. In addition to reporting the performance of each individual tool, we also include the results of model ensembles constructed from these tools. Because SAM 2 cannot directly process text queries, we provide it with GT bounding boxes and report its segmentation results accordingly. Notably, Ophiuchus surpasses both tools themselves, showing that the performance of these segmentation tools does not constitute a hard upper bound for our method. Through end-to-end training and optimization, Ophiuchus enhances the MLLM's self-reflection and decision-making abilities: it can detect when the segmentation output from one tool is unreliable and switch to another tool (see Figure 12). Moreover, Ophiuchus can compose multiple tools—such as combining the zoom-in tool with SAM 2—to further improve segmentation performance (see Figure 14). These experiments further underscore our contribution: by explicitly integrating tool usage into the model's own chain of thought, we strengthen the MLLM's visual perception, decision-making, and reasoning capabilities, enabling the emergence of higher-level abilities such as self-reflection and tool composition—capabilities absent in existing "think with images" approaches.

## D.5. Comparison Using the Same External Tools and Training Data

We provide in the Table 11 a comparable setting in which all LLMs and agents have access to the same external tools and are evaluated under the same training data conditions. Specifically, for close-source baselines, since these models cannot be further trained, we expose to them the full interfaces of all tools used in our system (zoom-in, SAM 2, and BioMedParse). We also explicitly instruct them in the prompt that they may invoke these tools for analysis. In addition, our evaluation code feeds the tool outputs back to these models for subsequent reasoning. For open-source baselines and agents, we not only provide unrestricted access to all tools but also fine-tune them using the complete three-stage training data we constructed with Gemini-2.5-Pro and GPT-5. This enables a fully equitable comparison. Even when evaluated with the same external tools and further fine-tuned on the same datasets, existing baseline methods still fall significantly short of our agent. This confirms that the performance gains of our model do not simply arise from using external tools or training data constructed by external models (Gemini-2.5-Pro and GPT-5). Instead, the improvement primarily comes from our proposed pixel-level visual reasoning paradigm, which explicitly integrates tool invocation and visual evidence into the model's chain-of-thought. This enables the model to learn when to use a tool and how to compose multiple tools for fine-grained visual analysis.

Importantly, this capability is far from a trivial form of tool use. During fine-tuning of existing baselines, we observed that these models generally lack sufficient understanding of complex vision-based tools. Because tool usage introduces external tokens and additional visual features, current methods struggle to interpret such information, failing to determine the appropriate timing for tool invocation and to effectively utilize the returned visual evidence. As a result, their performance may even degrade on certain test sets. This issue is especially pronounced in smaller-parameter models with weaker instruction-following abilities. Our proposed "think with images and tools" framework is specifically designed to strengthen an MLLM's comprehension of tools and visual feedback, thereby enabling effective tool reasoning and robust pixel-level visual understanding.

*Table 11.* Comparison across 8 VQA benchmarks using the same external tools configuration.

| Methods (same tools) | $D_{test}$-VQA | PathVQA | SLAKE | VQA-RAD | OmniMed | MMMU (H&M) | MedXpertQA | In-House-VQA | Avg |
|---|---|---|---|---|---|---|---|---|---|
| *Close-Source SOTA* | | | | | | | | | |
| GPT-4.1 | 36.9 | 58.9 | 72.4 | 65.5 | 75.4 | 73.1 | 41.3 | 25.8 | 58.6 |
| GPT-5 | 38.5 | 61.2 | 73.9 | 64.8 | 75.0 | 70.3 | 41.0 | 27.9 | 59.9 |
| OpenAI-o3 | 41.4 | 67.9 | 77.5 | 66.2 | 73.8 | 74.1 | 44.6 | 30.9 | 62.2 |
| Gemini 2.5 Pro | 41.9 | 68.2 | 73.6 | 64.2 | 77.2 | 72.3 | 46.9 | 30.5 | 61.8 |
| *Open-Source SOTA* | | | | | | | | | |
| InternVL3-8B | 39.7 | 54.1 | 71.0 | 66.4 | 72.5 | 62.7 | 24.0 | 21.1 | 49.0 |
| LLaVA-Next-13B | 23.5 | 40.3 | 57.9 | 55.2 | 58.4 | 39.2 | 19.4 | 18.2 | 41.4 |
| Qwen2.5-VL-32B | 38.7 | 48.9 | 70.8 | 71.9 | 69.8 | 61.2 | 26.9 | 26.1 | 53.7 |
| *MLLMs can "Think with Images"* | | | | | | | | | |
| DeepEyes-7B | 40.3 | 53.4 | 68.7 | 66.3 | 65.4 | 58.2 | 23.9 | 21.3 | 50.3 |
| Mini-o3-7B-v1 | 40.7 | 53.8 | 68.6 | 66.1 | 65.9 | 57.7 | 24.4 | 22.1 | 50.3 |
| PixelReasoner-RL-v1-7B | 40.9 | 54.2 | 68.4 | 66.5 | 65.4 | 58.4 | 23.8 | 22.0 | 50.4 |
| *Medical MLLMs* | | | | | | | | | |
| LLaVA-Med-7B | 27.5 | 45.0 | 47.9 | 52.7 | 52.3 | 38.5 | 19.4 | 19.2 | 39.3 |
| HuatuoGPT-Vision-34B | 37.9 | 51.3 | 68.5 | 62.0 | 70.9 | 59.5 | 24.2 | 21.7 | 51.5 |
| *Medical MLLMs with CoT Reasoning* | | | | | | | | | |
| Med-R1-2B* | 18.3 | 19.6 | 52.3 | 36.4 | – | 44.5 | 22.6 | 16.3 | – |
| Lingshu-7B | 41.1 | 69.8 | 78.5 | 66.9 | 75.6 | 58.4 | 26.5 | 26.3 | 57.4 |
| Chiron-o1-8B | 41.3 | 70.3 | 78.2 | 72.8 | 77.0 | 56.7 | 25.9 | 28.4 | 58.2 |
| *Multimodal Medical Agents* | | | | | | | | | |
| MMedAgent-7B | 40.6 | 61.5 | 69.5 | 64.8 | 58.9 | 45.0 | 23.3 | 20.8 | 49.4 |
| MMedAgent-RL-7B | 39.5 | 60.4 | 68.1 | 66.3 | 61.4 | 59.9 | 23.5 | 23.9 | 50.6 |
| VILA-M3-40B | 41.2 | 68.7 | 73.9 | 67.7 | 57.5 | 57.6 | 24.1 | 28.3 | 50.0 |
| Qwen2.5-VL-7B | 36.2 | 53.1 | 67.7 | 64.5 | 65.3 | 57.8 | 24.7 | 20.4 | 47.0 |
| **Ophiuchus-7B (Ours)** | **59.4** | **74.3** | **83.9** | **73.6** | **78.6** | **76.0** | **39.3** | **50.4** | **68.0** |

*Table 12.* Analysis of Over-Calling of Tools.

| Benchmark | Percentage of Excessive Tool-Calling Behavior (%) |
|---|---|
| $D_{test}$-VQA (In-domain) | 4.8 |
| PathVQA | 1.4 |
| SLAKE | 4.2 |
| VQA-RAD | 0.6 |
| OmniMed | 3.5 |
| MMMU (H&M) | 3.7 |
| MedXpertQA | 3.9 |
| In-House-VQA | 4.6 |

## D.6. Analysis of Over-Calling of Tools.

We also include a small subset of questions (3K) that GPT-5 can correctly answer without tools during training, with the goal of teaching the model to invoke tools only when necessary. As a result, our agent directly answers questions that it considers sufficiently simple or solvable using the MLLM's innate capabilities. We provide an analysis of over-calling of tools in Table 12. Specifically, we first identify, for each benchmark, the set of cases that the base model Qwen2.5-VL-7B can answer correctly without using any tools. We then count how many of these cases Ophiuchus chooses to answer using tools. Finally, we report the proportion of such "directly answerable" cases for which Ophiuchus still decides to use a tool. The results show that for questions the base model can already answer correctly, our agent rarely performs additional tool calls, thereby improving performance while maintaining efficiency.

## D.7. Stability Analysis of Training.

In Figure 16, we present the training curves, including the SFT training-loss curve, the RL overall-reward curve, the reasoning-format reward curve, the strategic tool-use reward curve, and the IoU curve of the final-answer reward. These curves demonstrate that the SFT and RL training dynamics of Ophiuchus exhibit stable and consistent improvements.

we also adjusted the RL hyperparameters and evaluated the resulting performance changes to assess the robustness of

*Table 13.* Ablation study on RL hyperparameters (learning rate lr and number of CoT rollouts $N_c$) evaluated on 8 VQA benchmarks. Default setting is highlighted in bold.

| RL Hyperparameters | D_test-VQA | PathVQA | SLAKE | VQA-RAD | OmniMed | MMMU (H&M) | MedXpertQA | In-House-VQA |
|---|---|---|---|---|---|---|---|---|
| lr $= 1 \times 10^{-6}$, $N_c = 4$ | 59.1 | 74.0 | 83.3 | 73.5 | 78.6 | 75.4 | 39.1 | 50.2 |
| lr $= 1 \times 10^{-6}$, $N_c = 8$ (default) | **59.4** | **74.3** | **83.9** | **73.6** | 78.6 | **76.0** | **39.3** | **50.4** |
| lr $= 1 \times 10^{-5}$, $N_c = 4$ | 59.0 | 74.2 | 83.5 | 73.1 | 78.7 | 75.8 | 39.0 | 50.1 |
| lr $= 2 \times 10^{-5}$, $N_c = 8$ | 59.2 | 74.3 | 83.6 | 73.4 | 78.7 | 76.0 | 39.2 | 50.0 |
| lr $= 2 \times 10^{-6}$, $N_c = 8$ | 59.3 | 74.3 | 83.8 | 73.5 | 78.6 | 75.9 | 39.2 | 50.3 |

*Table 14.* Comparison of finding generation performance across multiple models.

| Models (Finding Generation) | BERTScore | BLEU | ROUGE-1 | ROUGE-2 | ROUGE-L |
|---|---|---|---|---|---|
| HuatuoGPT-Vision-7B | 85.4 | 9.7 | 27.8 | 5.3 | 16.1 |
| Qwen2.5-VL-72B-Instruct | 84.6 | 5.6 | 22.7 | 4.2 | 14.2 |
| CheXagent (Chen et al., 2024c) | 74.6 | 0.3 | 4.7 | 0.1 | 4.3 |
| ChestX-Reasoner (Fan et al., 2025) | 82.2 | 4.8 | 14.4 | 4.1 | 11.5 |
| Qwen2.5-VL-7B-Instruct | 84.5 | 7.2 | 22.4 | 4.2 | 14.0 |
| GPT-4o | 86.2 | 11.1 | 30.4 | 6.4 | 18.4 |
| GPT-4o-mini | 85.2 | 6.7 | 22.8 | 3.8 | 14.2 |
| DeepSeek-VL2 | 82.3 | 4.4 | 14.9 | 3.0 | 10.6 |
| Chiron-o1-8B | 86.2 | 10.7 | 26.7 | 7.4 | 17.8 |
| **Ophiuchus-7B (ours)** | **87.9** | **13.5** | **32.9** | **8.6** | **18.3** |

Ophiuchus. Specifically, we varied the learning rates $lr$ and the number of roll-out candidate reasoning paths $N_c$. The detailed results are summarized in the Table 13. The results show that adjustments to the RL hyperparameters lead to only minor performance fluctuations. Overall, our default parameter settings achieve the best performance, indicating that our choices of RL hyperparameters are appropriate and well justified.

### D.8. Generalization to Report Generation Tasks.

We adopt MIMIC-CXR (Johnson et al., 2019) for chest X-ray report generation, covering two tasks: finding generation and impression generation. We evaluated Ophiuchus's performance using BERTScore, BLEU, ROUGE-1, ROUGE-2, and ROUGE-L as metrics. The results for both tasks are shown in Table 14 and Table 15, respectively. It is evident that Ophiuchus achieves superior performance compared with both general-purpose large-scale MLLMs and existing medical MLLMs. This improvement can be attributed to Ophiuchus's enhanced ability to understand and identify fine-grained visual cues—capabilities that are also essential for accurate and clinically meaningful report generation.

## E. Case Studies

### E.1. Successful Emergent Thinking Patterns

We identify three primary patterns.

**(1) Visual-Cue Search.** When a single global observation of the image is insufficient for a complex problem, the model leverages segmentation tools to generate region proposals and employs a zoom-in tool to examine them, aggregating visual cues and reasoning over them to reach a reliable conclusion.

As shown in Figure 8, **Ophiuchus** begins from the full endoscopic view and judges that a global look is insufficient for reliable counting. It first invokes BiomedParse to propose candidate polyp regions (Turn 1), then applies the zoom-in tool to each segmented area for close inspection of morphology and boundaries (Turn 2). Aggregating the zoomed evidence, it confirms two distinct polyps and outputs the correct answer (Turn 3), which exemplifies the visual-cue search pattern.

**(2) Visual Confirmation.** In some cases, the model starts uncertain but gradually builds confidence by analyzing tool-returned regional observations and zooming in on details to gather evidence and resolve ambiguities.

As shown in Figure 9, given an abdominal CT slice, **Ophiuchus** first performs anatomic elimination to down-weight pancreas, spleen, and renal options, yet remains uncertain about a suspected colonic wall finding. It therefore zooms into a targeted region of interest to verify the local appearance (Turn 1). The enlarged view reveals focal thickening and luminal irregularity (Turn 2), which provides direct confirmation and leads to the final choice. This illustrates how local evidence consolidates an initially uncertain hypothesis.

*Table 15.* Comparison of impression generation performance across multiple models.

| Models (Impression Generation) | BERTScore | BLEU | ROUGE-1 | ROUGE-2 | ROUGE-L |
|---|---|---|---|---|---|
| HuatuoGPT-Vision-7B | 83.5 | 2.8 | 10.6 | 1.8 | 7.5 |
| Qwen2.5-VL-72B-Instruct | 83.3 | 2.8 | 10.2 | 2.2 | 6.9 |
| CheXagent (Chen et al., 2024c) | 81.4 | 7.3 | 9.8 | 0.0 | 9.8 |
| ChestX-Reasoner (Fan et al., 2025) | 83.6 | 4.5 | 12.3 | 4.0 | 10.8 |
| Qwen2.5-VL-7B-Instruct | 83.6 | 3.7 | 10.0 | 2.0 | 7.2 |
| GPT-4o | 84.5 | 4.1 | 13.9 | 3.0 | 9.8 |
| GPT-4o-mini | 83.7 | 2.5 | 9.7 | 1.4 | 6.6 |
| DeepSeek-VL2 | 82.4 | 2.6 | 7.1 | 0.9 | 6.3 |
| Chiron-o1-8B | 83.8 | 3.7 | 10.3 | 2.0 | 8.1 |
| **Ophiuchus-7B (ours)** | **85.7** | **5.2** | **16.9** | **4.8** | **11.4** |

**(3) Hallucination Mitigation.** Although MLLMs can sometimes hallucinate, invoking the tools helps the model focus on visual details to mitigate hallucination and produce a more accurate diagnostic conclusion.

As shown in Figure 10, the initial free-view impression includes round morphology, hypodensity, and a tentative heterogeneous interior. To verify and avoid over-interpretation, **Ophiuchus** segments the suspected hepatic lesion with BiomedParse (Turn 1) and then zooms into the mask for careful inspection (Turn 2). The close view shows a well-circumscribed round hypodense mass with homogeneous attenuation, correcting the earlier hypothesis and yielding the final choices (Turn 3), which demonstrates hallucination mitigation through tool-guided re-examination.

### E.2. Failure Cases

We next analyze a representative hard case that illustrates both the limits and the introspective behavior of **Ophiuchus** (Figure 11). The image exhibits complex appearance with extremely subtle cues, and the pancreatic lesion occupies a tiny, hard-to-discern region. It first correctly interprets the user's goal-probe for possible pancreatic cancer-and, acknowledging uncertainty, **invokes the external segmentation tool BiomedParse with appropriate parameters.** The tool returns an incorrect mask that misses the true lesion. To validate the tool's output, it applies the zoom-in tool to the segmented area and inspects local morphology, **after which it rejects the tool hypothesis as inconsistent with pancreatic cancer.** Lacking reliable visual evidence, it ultimately predicts "no pancreatic cancer," which is wrong in this case. Although the final answer is incorrect and both the model and the tool fail under this ultra-low-signal setting, the trajectory evidences strong capability for intelligent tool invocation and reflective error checking: **Ophiuchus** localizes uncertainty, calls an appropriate tool, cross-examines the tool result via zoom-in verification, and explicitly diagnoses the tool's failure before committing to a conclusion.

### E.3. Self-Reflection Examples

As shown in Figure 12, **Ophiuchus** demonstrates explicit self-reflection and adaptive tool use. It first applies BiomedParse to detect a pancreatic cyst and inspects the returned mask. Based on anatomic reasoning, the model judges that the highlighted region corresponds to renal parenchyma rather than the pancreas, which signals a likely tool error. It then revisits the original CT, relocates the pancreas, formulates a revised hypothesis of a round hypodense lesion in the pancreatic head, and selects an alternative tool. It parameterizes SAM2 with a bounding box over the suspected lesion, obtains a precise segmentation, and confirms the cystic lesion. This trajectory shows **the model diagnosing a tool failure, switching tools, refining parameters, and verifying the final conclusion through visual evidence.**

## F. Prompt

### F.1. Prompt for VQA Construction and Verification

In this section, we present the prompt templates used for constructing VQA items and for verifying their quality. Figure 17 illustrates the prompt designed to generate VQA questions and answers from medical data, while Figure 18 shows the template used to check the quality and validity of the generated VQA pairs. These prompts are employed throughout data construction to ensure that the resulting dataset is both diverse and reliable. We follow the same template design when constructing segmentation-task data.

**F.2. Prompt for Reasoning Trajectory Construction**

In this section, we introduce the prompts for building reliable agentic reasoning trajectories for our curated VQA data. Unlike generic approaches that directly sample from a teacher model, we stabilize generation by providing GPT-5 with ground-truth clinical metadata and tool observations during prompting while explicitly instructing the agent to treat the answer as unknown throughout reasoning and reveal it only at the end. Using this prompt together with verified medical labels, we perform multi-turn rollout with environment interaction and tool use to construct step-by-step trajectories. Figure 19 presents the generation template, and Figure 20 provides the verification template that enforces strict checks on trace format, tool correctness, observation integration, clinical soundness, and exact answer match, with explicit focus on ROI/mask usage and zero tolerance for invented observations.

**F.3. System Prompt and User Prompt**

In this section, we present the two prompt templates used in our experiments. Figure 21 shows the System Prompt template. Figure 22 shows the User Prompt template.

# G. Limitations

First, a primary limitation is cost. Agentic RL with multi-turn rollouts, tool execution, and verifiable rewards is compute-intensive, and generating reasoning trajectories consumes substantial API resources.

Second, the present agent system remains 2D-centric, which limits performance on tasks that depend on 3D topology and motion.

# H. Future Work

We will broaden the agent's medically oriented toolset and extend the framework from 2D to volumetric and temporal modalities, including MRI, CT, and video.

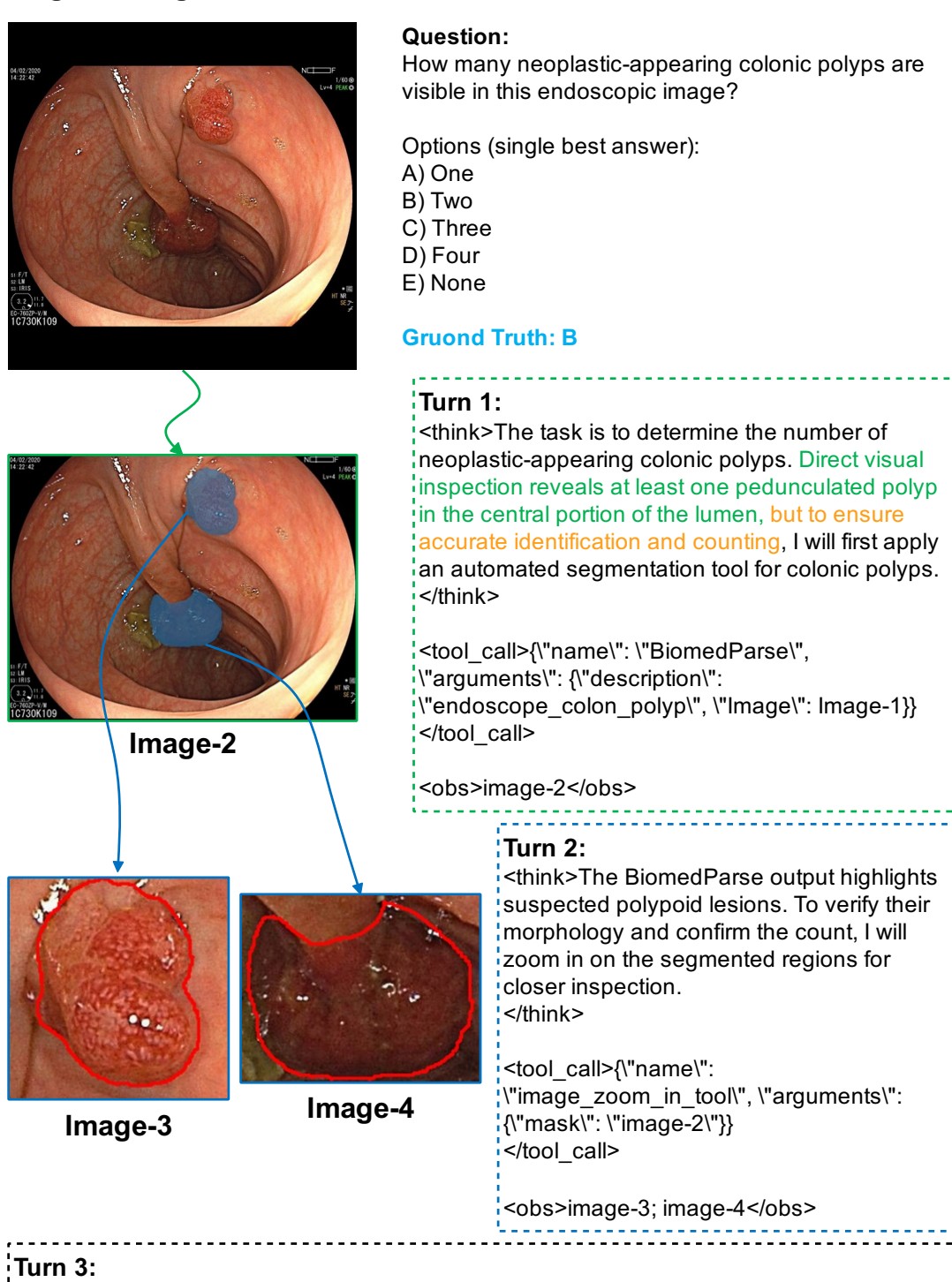

*Figure 8.* **Visual-Cue Search example.**

## Original Image-1:

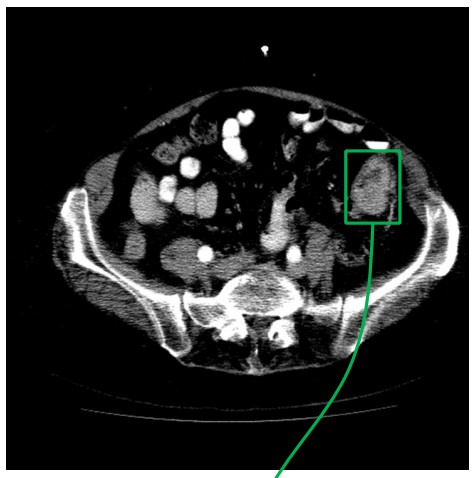

**Question:**
On this CT scan, which of the following best describes the abnormality (if any)?

Options (single best answer):
A) Pancreatic cancer
B) Splenic lesion
C) Renal tumor (renal neoplasm)
D) No abnormality detected
E) Colonic tumor (colorectal neoplasm)

**Gruond Truth: E**

**Turn 1:**
<think>Initial review of the abdominal CT at a lower abdominal–pelvic level. The pancreas is not visible at this slice, so option A) Pancreatic cancer is not applicable. The spleen is typically located in the left upper abdomen and is not included in this slice, so option B) Splenic lesion is excluded. Portions of both kidneys are visible, but there is no evidence of mass or focal lesion, so option C) Renal tumor is unlikely. In the left lower abdomen (right side of the image), bowel loops are visible with enhancing colonic wall. It is uncertain whether this represents normal enhancement or a possible abnormality. To clarify, I will apply a zoom-in tool to the region of interest.
</think>

<tool_call>{\"name\": \"image_zoom_in_tool\", \"arguments\": {\"bbox_2d\": [765, 331, 850, 458]}}
</tool_call>

<obs>image-2</obs>

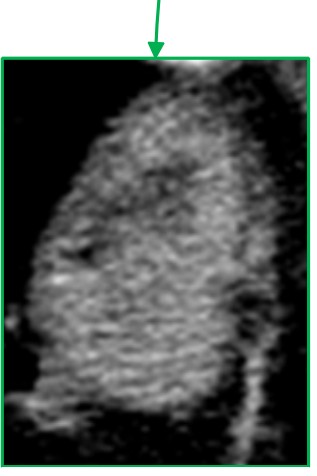

**Image-2**

**Turn 2:**
<think>The zoomed-in region shows a segment of colonic wall with focal thickening and luminal irregularity. These findings are consistent with an abnormal lesion rather than normal bowel wall. This supports option E) Colonic tumor (colorectal neoplasm) and rules out option D) No abnormality detected.
</think>

<answer>E</answer>

*Figure 9.* **Visual Confirmation example.**

**Original Image-1:**

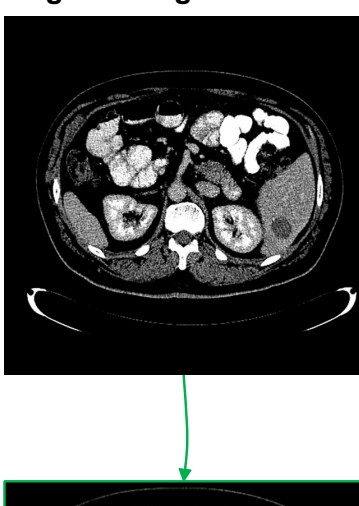

**Question:**
Which of the following descriptors apply to the hepatic lesion in this image?

Options (select all that apply):
A) Round-like morphology
B) Hypodense lesion
C) Heterogeneous internal intensity
D) Ill-defined margins
E) Well-defined margins

**Gruond Truth: ABE**

**Turn 1:**
<think>Initial inspection of the axial contrast-enhanced CT reveals a round-like lesion within the hepatic parenchyma. The lesion demonstrates attenuation lower than the surrounding liver tissue, indicating a hypodense character. Internally, the density appears relatively heterogeneous, and the margins are well defined, allowing clear demarcation from adjacent liver parenchyma. Based on the provided options, the preliminary answer would be A) Round-like morphology, B) Hypodense lesion, C) Heterogeneous internal intensity, and E) Well-defined margins. To confirm this assessment, I will apply BiomedParse with the description 'liver lesion' to obtain a precise segmentation.
</think>

<tool_call>{\"name\": \"BiomedParse\", \"arguments\": {\"description\": \"liver lesion\"; \"Image\": Image-1}}
</tool_call>

<obs>image-2</obs>

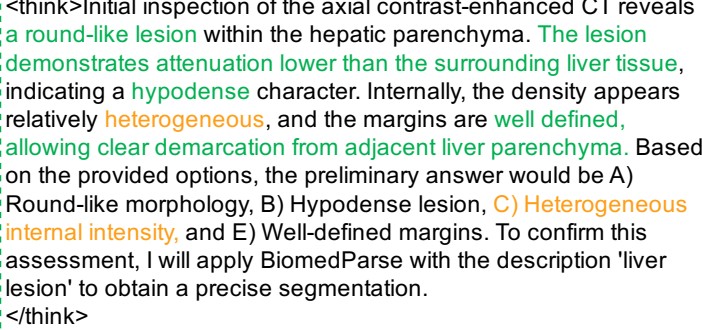

**Image-2**

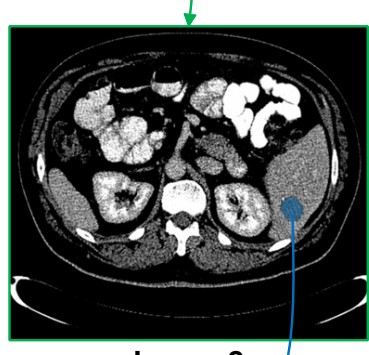

**Image-3**

**Turn 2:**
<think>The BiomedParse output provides a segmentation mask of the suspected hepatic lesion. To further refine my analysis, I will zoom in on the segmented region to evaluate its internal characteristics and margins more carefully.
</think>

<tool_call>{\"name\": \"image_zoom_in_tool\", \"arguments\": {\"mask\": \"image-2\"}}
</tool_call>

<obs>image-3</obs>

**Turn 3:**
<think>The zoomed-in image of the hepatic lesion confirms a well-circumscribed, round hypodense mass with homogeneous internal attenuation. My earlier impression of heterogeneous intensity was incorrect upon closer inspection. Therefore, the corrected final answer should include A) Round-like morphology, B) Hypodense lesion, and E) Well-defined margins.
</think>

<answer>ABE</answer>

*Figure 10.* **Hallucination Mitigation example.**

**Original Image-1:**

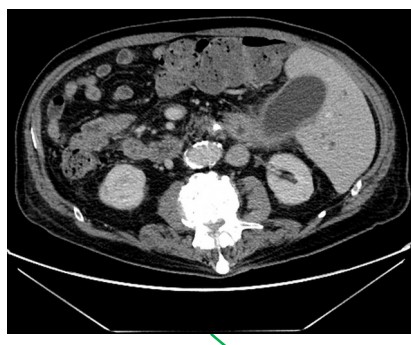

**Question:**
On the provided contrast-enhanced abdominal CT, is there evidence of pancreatic carcinoma? If present, delineate the tumor margins on the relevant slice(s).

Options:
A) Yes
B) No

**Gruond Truth: A) Yes**

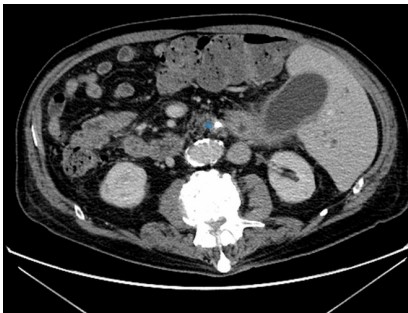

**Ground Truth** ✅

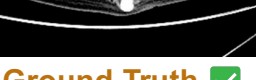

**Image-2** ❌

**Turn 1:**
<think>Initial review of the contrast-enhanced abdominal CT. The user's query is to determine whether a pancreatic carcinoma is present. On this slice, the pancreas is partially visible but there is no obvious mass, focal hypoattenuation, or infiltrative lesion. However, to confirm this impression, I will apply BiomedParse with the description 'pancreatic tumor'.</think>

<tool_call>{\"name\": \"BiomedParse\", \"arguments\": {\"description\": \"pancreatic tumor\", \"Image\": Image-1}}</tool_call>

<obs>image-2</obs>

**Turn 2:**
<think>The BiomedParse output highlights a region. To evaluate its validity, I will zoom in on the segmented area to assess whether it corresponds to a pancreatic mass.</think>

<tool_call>{\"name\": \"image_zoom_in_tool\", \"arguments\": {\"mask\": \"image-2\"}}</tool_call>

<obs>image-3</obs>

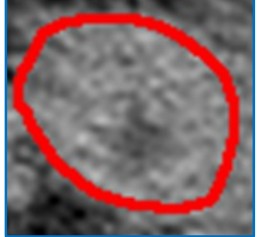

**Image-3**

**Turn 3:**
<think>The zoomed-in view shows that the segmented area does not correspond to a malignant pancreatic lesion. There is no mass effect, no irregular contour, and no hypoenhancing lesion consistent with pancreatic carcinoma. Therefore, the segmentation result is considered incorrect. Based on direct image review, this CT slice does not demonstrate pancreatic carcinoma.</think>

<answer>B</answer> 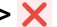 ❌

*Figure 11.* **Failure example (hard case).**

**Original Image-1:**

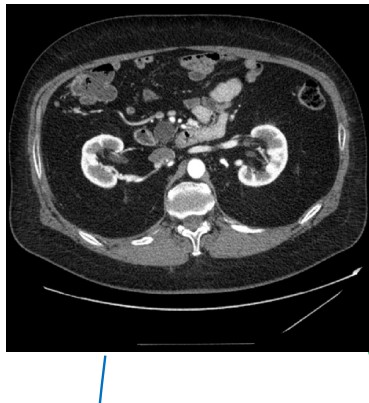

**Question:**
On the provided abdominal CT, is there evidence of a pancreatic cyst (pancreatic cystic lesion)? If present, delineate the lesion.

Options:
A) Yes
B) No

**Gruond Truth: A) Yes**

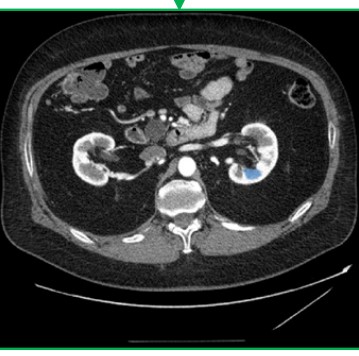

**Image-2**

**Turn 1:**
<think>Initial review of the contrast-enhanced abdominal CT. The task is to determine whether a pancreatic cystic lesion is present. To begin, I will apply an automated parsing tool focused on detecting pancreatic cysts.
</think>

<tool_call>{\"name\": \"BiomedParse\", \"arguments\": {\"description\": \"pancreatic cyst\", \"Image\": \"Image-1\"}}
</tool_call>

<obs>image-2</obs>

**Turn 2:**
<think>The BiomedParse output highlights a region. However, the segmentation appears to correspond to renal parenchyma rather than the pancreas. This suggests a misclassification. Therefore, I must reassess the original CT and localize the pancreas directly. Re-examining the original image-1, I observe a well-circumscribed, round, hypodense lesion in the region of the pancreatic head. This morphology is more consistent with a pancreatic cyst. To obtain a precise delineation, I will now apply the SAM2 model with a bounding box over the suspected lesion.
</think>

<tool_call>{\"name\": \"SAM2\", \"arguments\": {\"bbox_2d\": [554, 516, 612, 564], \"Image\": \"Image-1\"}}
</tool_call>

<obs>Image-3</obs>

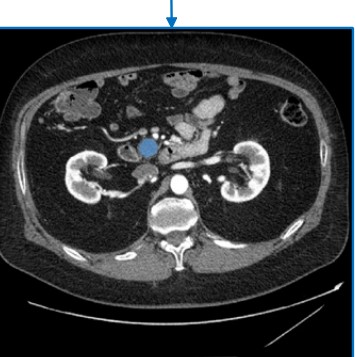

**Image-3**

**Turn 3:**
<think>The SAM2 segmentation outlines a round hypodense lesion in the pancreatic head, which confirms a cystic lesion with homogeneous internal attenuation and well-defined margins. This is consistent with a pancreatic cyst. Therefore, the correct answer is Yes, with the segmentation mask provided as mask-3.
<think>

**<answer>A, the segmentation mask is Image-3</answer>**

*Figure 12.* **Self-Reflection example.**

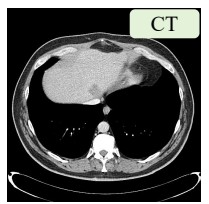

Question:

Does this CT image show any abnormalities? If there is any, please segment the abnormal area and provide a description of the anomaly.

Options:
A. Yes;
B. No;

GT Mask:

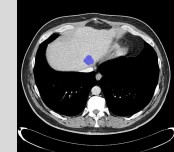

GT Answer:
A.
Yes, there is a hepatic tumor

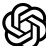 GPT-4o + Zoom-in

A. In the liver area, there is a low-density block ....., A bounding box is added in the figure to indicate the location of the lesion.

Incorrect Grounding 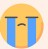

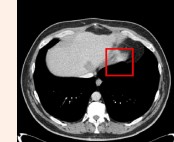

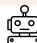 Lingshu-7B + SAM 2

A. There are multiple low-density lesion areas in the liver region. I generated bounding boxes for positioning and used SAM 2 for segmentation.

Incorrect reasoning analysis with 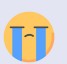
Incorrect Grounding

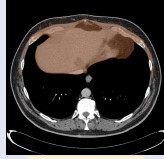

Dice: 3.29

**Ophiuchus-7B (ours)**

A. This Contrast-enhanced CT shows a roughly round, hypodense hepatic tumor in the liver area. To obtain a precise delineation, I will now apply the SAM2 model with a bounding box over the detected lesion.

Accurate spatial understanding 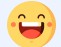
and High-quality masks !

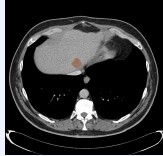

Dice: 98.42

*Figure 13.* **Comparative case analyses.** It can be observed that both general-domain and medical-specific agents augmented with tools frequently produce incorrect interpretations and answers when analyzing medical images, often accompanied by inaccurate localizations or segmentation masks. In contrast, Ophiuchus not only produces correct textual analyses but also achieves accurate spatial understanding, generating high-quality segmentation masks.

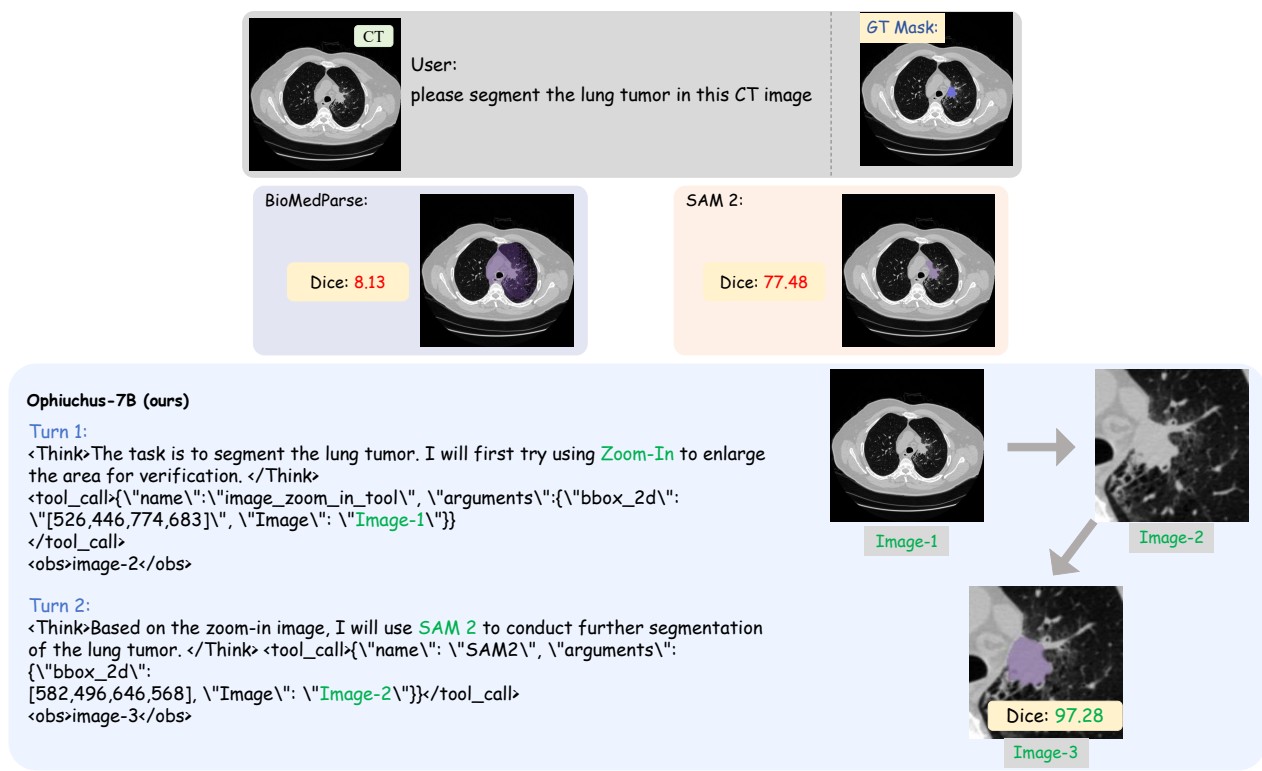

*Figure 14.* **Cases illustrating how Ophiuchus combines different tools to further enhance segmentation performance.** Ophiuchus can compose multiple tools—by using the zoom-in tool in conjunction with SAM 2—to achieve better segmentation results than relying on SAM 2 alone.

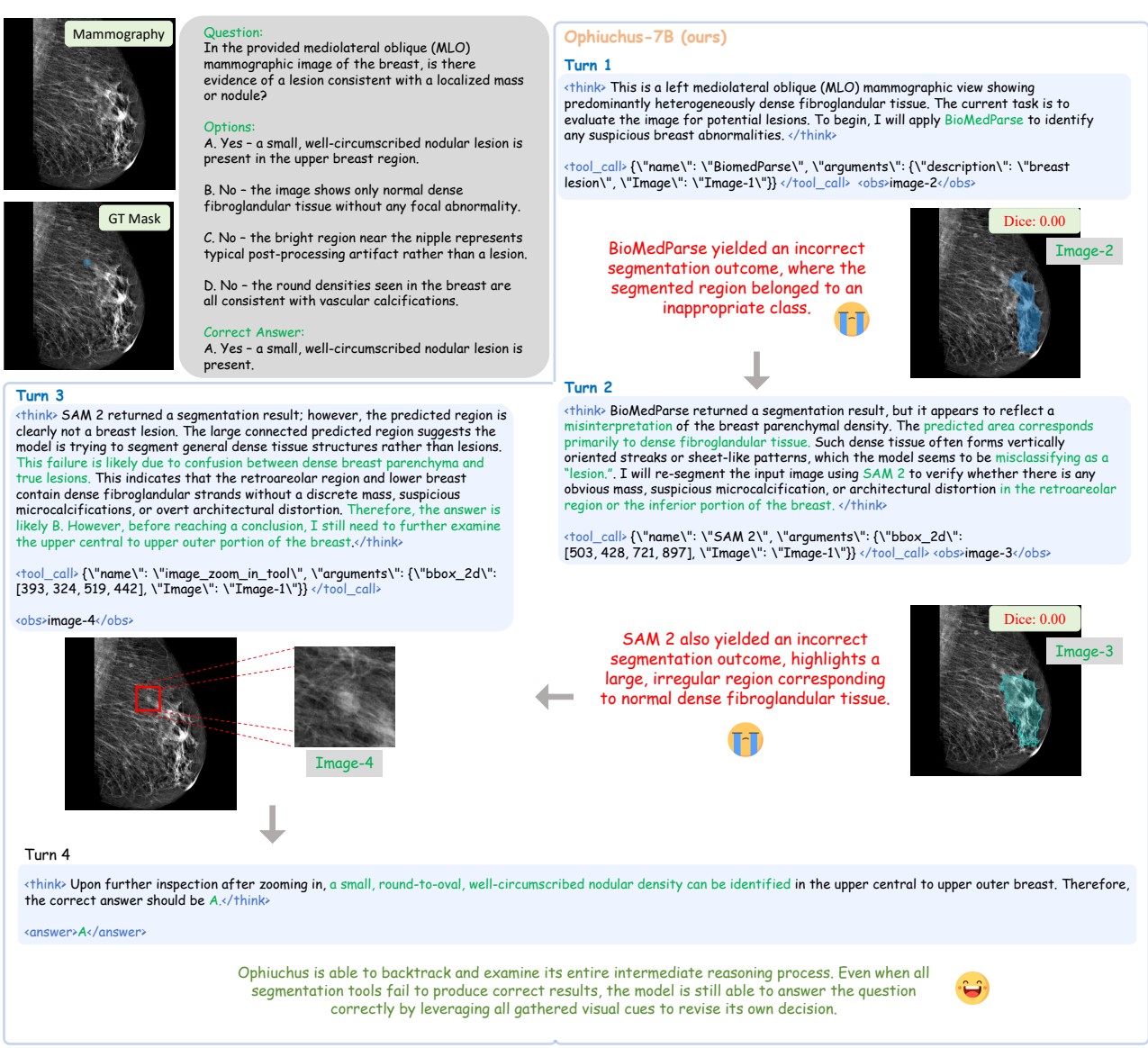

*Figure 15.* **Cases illustrating that even when all segmentation tools fail, Ophiuchus can still answer the question correctly by leveraging all gathered visual cues to revise its own decision.**

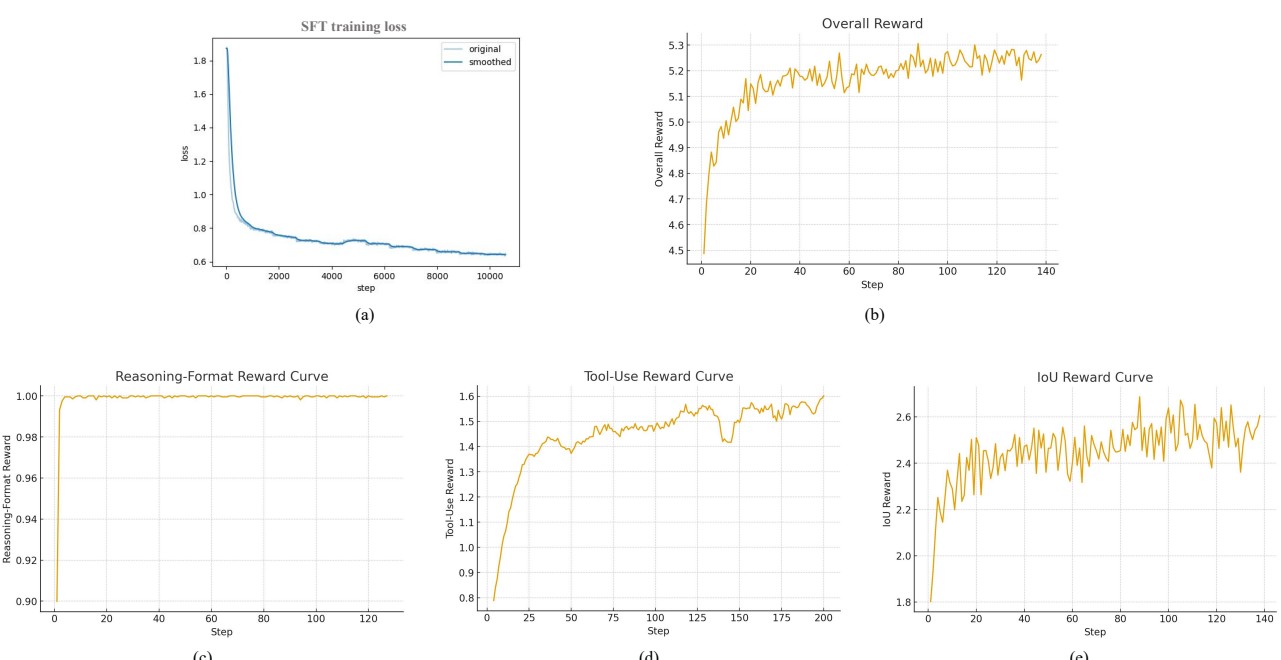

*Figure 16.* **Training curves of Ophiuchus. (a) Training loss during the SFT stage; (b) overall reward curve during RL training; (c) reasoning-format reward curve; (d) strategic tool-use reward curve; (e) IoU-based component of the final-answer reward.**

You are a medical expert specializing in interpreting medical images. Your task is to generate a clinically relevant visual multiple-choice question-and-answer set based on the provided inputs. The user will supply the following:

- Medical image: 1024 × 1024 resolution
- Target region coordinates: in the format <"bbox": [x1, y1, x2, y2]>, where (x1, y1) is the top-left corner and (x2, y2) is the bottom-right corner
- Binary ground-truth mask: 1024 × 1024 resolution, indicating the target region
- Textual descriptions: medical descriptions of the target segmentation region

**Your task:**
1. Create one multiple-choice question that is directly answerable by analyzing the given image and the specified target region.
2. Provide four or five options (A–D or A–E), using clear and professional medical terminology.
3. Indicate the correct answer explicitly.
4. Return the result strictly in the following JSON format:

```
[{
  "Question": "xxx",
  "Options": {
    "A": "xxx",
    "B": "xxx",
    "C": "xxx",
    "D": "xxx"
  },
  "Answer": "A"
}]
```

**Notes:**
- The question must be independent and self-contained; avoid referential terms such as "this case" or "the image above". Each question should stand alone.
- Only use the information provided (image, bbox, mask, and descriptions). Do not hallucinate or introduce extra details.
- Use precise, standardized clinical language to maintain professional accuracy and clarity.
- The question should be strongly correlated with the target segmentation region and clinically meaningful.

Below are some examples of questions and answers for other images:

{seed_example_1}
{seed_example_2}

Now, use the following image, bbox, mask, and the mask descriptions to generate the question:

*Figure 17.* **Prompt for VQA construction.** Template for generating clinically grounded question-answer pairs under strict format control.

You are a medical imaging question–answer (VQA) validation expert. Validate a multiple-choice QA item using the provided image, target region (bbox + mask), and mask descriptions.

**Inputs**
1. Image: {image}
2. Image Resolution: {resolution} (expected: 1024×1024)
3. Target RegionBBox:
    - {bbox} in the format ["bbox": [x1, y1, x2, y2]] (top-left = (x1,y1), bottom-right = (x2,y2))
    - Binary Mask: {mask} (1024×1024)
4. Mask Descriptions: {mask_descriptions} (textual descriptors of the segmented region)
5. Generated QA JSON: {qa_json}
    - Expected schema:
      ```
      [{
        "Question": "string",
        "Options": { "A": "string", "B": "string", "C": "string", "D": "string", "E": "string (optional)" },
        "Answer": "A/B/C/D/E"
      }]
      ```

**Validation Criteria**
Evaluate only using the provided inputs. Do not introduce external knowledge or patient context that isn't present.
1. Schema & Format
    - JSON is valid and parseable.
    - Keys Question, Options, Answer exist.
    - Options contains 4 or 5 choices labeled A–D (and optionally E) only.
    - Answer is exactly a single uppercase letter among the provided options (A/B/C/D/E).
2. Independence & Wording
    - The question is self-contained; avoids referential terms.
    - Professional, precise clinical terminology; no vague or colloquial phrasing.
3. Grounding to Target Region
    - The question is directly answerable from the image with attention to the segmented target region (use the bbox and mask to focus reasoning).
    - High correlation with the target region: content should pertain to structures/findings inside or tightly adjacent to the masked area.
    - Uses only information derivable from the image + mask + mask descriptions; no hallucinated modality, demographics, or history unless explicitly given.
4. Options Quality
    - Options are mutually exclusive, non-overlapping, and plausible; one best answer.
    - No trivial cues (e.g., "All of the above," "None of the above"), and no duplicate or near-duplicate options.
5. Answer Correctness
    - Determine the ground-truth choice by analyzing the image with the bbox/mask focus and the mask descriptions.
    - Verify the provided Answer matches the correct option.

**Output (STRICT)**
Return exactly one of the following:

- If JSON/schema/options/answer format is invalid (including wrong key names, missing fields, invalid labels, or Answer not a single A/B/C/D/E), respond with: Format error: <brief reason>
- If the format is valid and the specified Answer is correct, respond only with: Correct.
- If the format is valid but the specified Answer is wrong, respond with: Incorrect: <brief explanation grounded in bbox/mask/descriptions>
- Keep the explanation concise, e.g., one sentence referencing the segmented region and salient imaging cues.

**Additional Guidance for Your Reasoning (do not output this text)**
- Use the bbox and mask to prioritize features within the target region; if the question ignores the region or can't be answered from it, treat as a format error ("question not grounded to target region").

- Prefer standardized terms (e.g., "hypodense lesion," "spiculated nodule," "vascular calcification") consistent with the mask descriptions.

- If the resolution is not 1024×1024, you may still judge correctness but do not infer missing details from the discrepancy; rely on what's given.

*Figure 18.* **Prompt for VQA Verification.** Template for assessing the quality of each generated VQA item.

You are a medical expert specializing in interpreting medical images. The user will provide you with the following inputs:
- A medical image (with metadata such as modality and view) and resolution
- An ROI defined by a bounding box (bbox)
- A segmentation mask and a textual description of the mask/target
- A multiple-choice question related to the image
- A list of answer options
- The ground-truth answer
- Precomputed tool observation blocks that will appear in context after each tool call, formatted as <obs> ... </obs>

You also have access to the following tools (you must call them using the specified format):
- image_zoom_in_tool — zooms into a region
  - Call format inside <tool_call>:
    {"name":"image_zoom_in_tool","arguments":{"bbox_2d":[x1,y1,x2,y2]} }
    (Optionally, you may pass a mask field instead of bbox_2d to zoom by mask.)

- SAM2 — segments a target given a bbox
  - Call format: {"name":"SAM2","arguments":{"bbox_2d":[x1,y1,x2,y2]}}

- BioMedParse — segments a target given a textual description
  - Call format: {"name":"BiomedParse","arguments":{"description":"<object/type description>"}}

Your task is to simulate a step-by-step diagnostic reasoning process that uses these tools and their provided <obs> results to reach the answer. You must assume you do not know the ground-truth answer in advance; do not reference it during the reasoning. After the reasoning completes, reveal the correct answer by copying the ground-truth answer exactly. Please do not give empty output.
- Carefully analyze the image description/metadata, ROI (bbox), mask + mask description, and the question.
- Plan which tool to call, why you need it, and what arguments to pass.
- Every tool call must be:
  - preceded by a <think> step that justifies the call and arguments, and
  - immediately followed by the corresponding <obs> block that is already provided in the context. Do not fabricate or alter <obs> contents; only use the given ones.
- Integrate each <obs> into the next <think> step; refine or backtrack if needed.
- Use concise, precise, and clinically sound language grounded only in the given information.
- End with an <answer> block.

**Output in JSON Format:**

{
  "trace": "<think>...</think>\n<tool_call>{...}</tool_call>\n<obs>...</obs>\n...\n<think>...</think>\nAction:
Answer\n<answer>...</answer>",
  "answer": "<groundtruth answer>"
}

Context Placeholders to be provided by the user:

- Image/Resolution: {image_metadata} / {resolution}
- BBox: {bbox}
- Mask: {mask}
- Mask Description: {mask_descriptions}
- Question: {question}
- Options: {options}
- Groundtruth answer: {answer}
- Tool Observation Blocks (precomputed, to be consumed after each tool call): {obs_blocks}

*Figure 19.* **Prompt for Reasoning Trajectory Generation.** Template for constructing reliable agentic reasoning trajectories.

You are a medical expert specializing in interpreting medical images. The user will provide:
• The full generated reasoning trace using <think>, <tool_call>, <obs>, and <answer>
• The original image metadata and resolution
• The ROI bbox, segmentation mask, and mask description
• The multiple-choice question, options, and the ground-truth answer

Your task is to strictly evaluate the quality of the reasoning trace and its tool use. Check format, tool correctness, integration of observations, clinical soundness, and final answer correctness. Base your evaluation only on the provided trace and context. Do not assume extra clinical information.

**Evaluation rules:**
• Format: The trace must interleave <think> → <tool_call> → <obs> steps (repeated as needed) and conclude with Action: Answer and an <answer> block.
• Tool usage: Only the allowed tools (image_zoom_in_tool, SAM2, BioMedParse) may be used with valid JSON arguments.
• Tool choice rationale: Each <tool_call> must be justified by the immediately preceding <think> (why this tool, why these arguments).
• Observation integration: Each <obs> must be meaningfully used in the subsequent <think>; no ignoring or contradicting tool outputs.
• Clinical correctness: Reasoning must be consistent with the image/ROI/mask and mask description; no hallucinated findings or unsupported claims.
• Final answer: The content of <answer> must match the provided ground-truth answer exactly.

**Output in JSON Format:**

{
 "format_check": "pass/fail",
 "tool_usage_check": "pass/fail",
 "tool_choice_rationale_check": "pass/fail",
 "obs_integration_check": "pass/fail",
 "clinical_correctness": "pass/fail",
 "final_answer_check": "correct/incorrect",
 "explanation": "Brief reason(s) for any fail/incorrect; empty if all pass."
}

**Notes:**

• Focus on the segmentation mask and bounding box details: If the problem provided a mask or ROI (region of interest), ensure the reasoning trace made use of it. Not using an important provided region could be a flaw (either in tool usage or obs integration).

• No credit for invented observations: If the reasoning trace mentions observations that were not actually present in the <obs> outputs, that is a mistake (either a hallucination in the reasoning or misuse of a tool). The evaluator should catch this under clinical correctness or tool usage (since the model claimed a tool result that wasn't real).

• Penalize redundant or unmotivated tool calls: If the chain-of-thought uses a tool when it's not necessary (for example, calling a zoom tool on a region that was already clearly described, or repeating the same tool without new justification), that should be noted (likely affecting the tool_choice_rationale_check or tool_usage_check).

• The final JSON should be complete and well-formed. Even if multiple checks fail, the output must list each field. Only use the specified keys and values (no additional keys, and string values exactly as "pass", "fail", "correct", "incorrect").

*Figure 20.* **Prompt for Reasoning Trajectory Verification.** Template for validating the correctness and reliability of generated reasoning trajectories.

You are a helpful assistant specialized in visual medical image analysis. You have access to several tools that help you segment and examine medical images (e.g. highlighting lesions or tumors) to answer questions. Your task is to carefully analyze the image and question, use the tools step-by-step, and provide a well-reasoned final answer.

**Tools**

You may call one or more functions to assist with the user's query.

You are provided with the following function signatures within <tools></tools> XML tags:

<tools> [ { "type": "function", "function": { "name": "SAM2", "description": "Segment an object in the image given a bounding box, using the SAM2 model for precise segmentation.", "parameters": { "type": "object", "properties": { "index": { "type": "number", "description": "The index of the image to operate on (if multiple images are present)." }, "bbox_2d": { "type": "array", "items": { "type": "number" }, "minItems": 4, "maxItems": 4, "description": "The bounding box of the target region to segment, as [x1, y1, x2, y2] (top-left and bottom-right corner coordinates)." }, "label": { "type": "string", "description": "Optional descriptor of the object to segment (e.g. 'liver tumor'), for logging or clarification." } }, "required": [ "bbox_2d" ] } } }, { "type": "function", "function": { "name": "BiomedParse", "description": "Detect and segment a specified object type in the image (e.g. lesion, tumor) using a biomedical image parser model.", "parameters": { "type": "object", "properties": { "index": { "type": "number", "description": "The index of the image to analyze." }, "object_type": { "type": "string", "description": "The type or label of the object to segment (e.g. 'liver tumor', 'liver')." } }, "required": [ "object_type" ] } } }, { "type": "function", "function": { "name": "image_zoom_in_tool", "description": "Zoom in on a specific region of an image by cropping it to a bounding box or mask, for detailed inspection. If a mask is provided, the zoomed image will highlight the mask's contour.", "parameters": { "type": "object", "properties": { "bbox_2d": { "type": "array", "items": { "type": "number" }, "minItems": 4, "maxItems": 4, "description": "The bounding box of the region to zoom in on, as [x1, y1, x2, y2]. Coordinates use the image's pixel space." }, "mask": { "type": "string", "description": "*(Optional)* A segmentation mask of the target object/region (as an image or reference). If provided, the zoomed-in view will outline this mask." }, "label": { "type": "string", "description": "*(Optional)* Name/label of the object in the specified region, for clarity." } }, "required": [ "bbox_2d" ] } } ] </tools>

**Tool Usage**

To use a tool, return a JSON object with the function name and arguments within <tool_call></tool_call> tags, for example:

<tool_call>
{"name": "image_zoom_in_tool", "arguments": {"index": 0, "bbox_2d": [50, 120, 200, 300]}}
</tool_call>

This will invoke the image_zoom_in_tool on the specified region of image 0. The results of the tool (e.g. a zoomed-in image) will then be appended to the conversation for you to examine in the next step.

**Important**: After each tool call, wait for the tool's output and analyze it before proceeding. The tool's result will appear in the context (for example, an image with a highlighted area). Use this feedback to decide your next step.

**Reasoning Steps and Output Format**

1. **Initial Analysis**: Begin by examining the question and image carefully. Break down the problem: what is being asked, and what information or measurements might be needed from the image? Plan an approach before using any tool.
2. **Iterative Tool Reasoning**: Tackle the task step-by-step using the tools:
   - **Choose the appropriate tool** for the current sub-task (e.g. use BiomedParse to detect and segment a tumor by specifying the object type, use SAM2 if you have or need a specific bounding box focus for segmentation, or image_zoom_in_tool to get a closer look at a region of interest).
   - **Form the tool call with proper arguments in JSON**:
     - If using coordinates (for segmentation or zoom), provide absolute pixel coordinates. Remember that the top-left corner of the image is (0,0) and the bottom-right is *(width, height)*. Ensure [*x1, y1*] is the top-left and [*x2, y2*] the bottom-right of the region.
     - If using BiomedParse, provide the *object_type* (e.g. "liver tumor" or "polyp in endoscopy") relevant to the question.
     - Double-check the JSON format and required fields for the tool.
   - Example (reasoning + tool call):
     <think> The question asks for the shape of a pancreas tumor in the CT scan. I will first segment the pancreas tumor. </think>
     <tool_call>{"name": "BiomedParse", "arguments": {"index": 0, "object_type": "pancreas tumor"}}</tool_call>
     (This will attempt to segment any pancreas tumor in image 0.)
   - **Observe the Tool Output:** Once the tool's result is returned (e.g. a mask outlining the tumor or a zoomed image of a region), examine it. Reflect on the visualization: Did the tool correctly identify the intended area? Does the segmentation mask accurately cover the lesion or tumor in question? If you used zoom, did it focus on the right region?
   - **Adjust if Necessary**: If the result is not accurate or complete, adjust your approach:
     - You may refine the bounding box and call SAM2 on a specific region if BiomedParse's result was too broad or missed something.
     - If the mask is incomplete or includes extra areas, consider calling the tool again with a corrected region or different parameters.
     - Backtrack to a previous step if needed. For example, if the first segmentation attempt fails to identify the tumor, reconsider if the correct object type or region was specified, and try again.
   - **Iterate**: Continue using tools in this manner (analyze -> tool call -> review output) until you have gathered enough information to answer the question confidently. Each tool usage should be accompanied by a <think> reasoning step explaining why you are using it and what you expect to learn.
3. **Formulating the Final Answer**: Once you have all necessary information (for example, the tumor is segmented and you've derived the size or other required insights from the image), formulate the final answer.
   - End your reasoning chain with an Answer action. Structure the final step as: <answer>[Your final answer to the user's question]</answer>
   - The content inside <answer> should directly address the user's question in a clear and concise manner, using the insights gained from the image analysis. This is the response the user will see.

**Note**: Throughout this process, maintain a clear and logical reasoning flow. Use the tools only as needed and do not include irrelevant information. The goal is to provide a correct, well-supported answer to the visual medical question, with the help of the tools to ensure accuracy in interpreting the image. Always double-check that the final answer is consistent with the visual evidence obtained from the tools.

*Figure 21.* **System Prompt.**

**Dialogue History**

<user> {{ Initial User Input }} </user>
<think> Round 1 Model Thought </think>
{{ Round 1 model output <tool_call> }}
<obs> Round 1 Observation </obs>
**... ...**

**Question**: {}

The index of the given image is {current_image_idx} (width: {width}, height: {height}). Continue your reasoning. After each tool use, critically evaluate the results and adjust if needed. When you believe that you have gathered sufficient information, please answer the initial question and structure your response as required.

*Figure 22.* **User Prompt**

