# OpenReview forum: "Ophiuchus: Incentivizing Tool-augmented ''Think with Images'' for Joint Medical Segmentation, Understanding and Reasoning"
_ICML.cc/2026/Conference — ICML 2026 regular_

### Official Review · Reviewer_ezX1 · 2026-02-21

**Soundness:** 3
**Presentation:** 3
**Significance:** 2
**Originality:** 2
**Overall Recommendation:** 4
**Confidence:** 5

**Summary:**

This work presents Ophiuchus, which is a tool-augmented framework for medical MLLMs that enables dynamic, fine-grained visual reasoning. By integrating external tools like Segment Anything Model 2 and BiomedParse into the reasoning loop, it lets the model to decide when and where to probe images and incorporates the results into multimodal chains of thought. Trained via a three-stage strategy, Ophiuchus is claimed to significantly outperform prior state-of-the-art models across multiple medical benchmarks, including VQA, detection, and segmentation.

**Compliance With Llm Reviewing Policy:**

Affirmed.

**Final Justification:**

Thanks the authors for their feedbacks. While I believe additional ablation studies and analyses would strengthen the paper by more systematically identifying the sources of performance gains, I agree that the work represents a valuable contribution. Accordingly, I am inclined to raise my score to Borderline Accept.

**Key Questions For Authors:**

1. Could you provide more evidence and justifications to explain what's the source of the significant improvements
2. What's the computational overhead
3. How to validate the accuracy and reliability of the constructed dataset
4. Can the framework integrate other tools which could contribute significantly to specialized tasks, like pathology examination

**Limitations:**

please see Weaknesses

**Strengths And Weaknesses:**

Strengths:

1. The topic is timely and of broad interest to the ICML audience.

2. The proposed method is technically sound and well-formulated.


Weaknesses:

1. Although the paper claims substantial improvements over existing methods, including 40B models, it is unclear where these gains originate. The ablation studies suggest that key components, such as tool reward design and tool selection, have limited impact on performance, leaving the source of the reported improvements somewhat ambiguous.

2. Relatedly, Table 3 and lines 399–402 indicate that even with ground-truth region crops, performance remains roughly 20% below the proposed model. This is confusing, as the external tools used (SAM2 and BiomedParse) are primarily segmentation models.

3. The additional computational cost of the iterative reasoning and refinement process is not quantified, making it difficult to assess efficiency and scalability.

4. The set of tools considered is quite limited, particularly for pathology tasks where the spatial distribution of different cell and tissue types is critical for clinical decisions. SAM2 and BiomedParse are known to struggle with dense or clustered structures.

5. There are concerns about the constructed dataset, as the QA pairs were generated by Gemini and do not appear to be validated by clinical experts.

---

> ### Author Rebuttal · Authors · 2026-03-30
>
> We thank the reviewer for recognizing the timeliness of our topic and the technical soundness of our method.
>
> >**W1&Q1**. More justifications on the source of the significant improvements.
>
> **R1**: Thank you for raising this important question. The performance gain comes not from tool use alone, but from the **pixel-level visual reasoning** ability elicited through tool integration. This ability enables the model to autonomously select tools, flexibly compose tool use without predefined workflows, and internalize tool outputs into its reasoning process. To elicit this capability, we design a progressive framework including the iterative thought–tool–observation loop, cold-start SFT, Self-Reflection Fine-Tuning, and Agentic RL with tool rewards. As shown in Table 2, all components contribute meaningfully, indicating that the improvement comes from their synergy rather than any single component. Tool reward is intended to make tool invocation more meaningful, but it is not the sole source of the overall improvement.
>
> Moreover, the key factor is less which tools are selected than how they are used. As shown in Supplementary Fig. 13, existing tool-augmented MLLMs often provide inaccurate tool parameters, causing tool use to introduce noise rather than help. Supplementary Table 6 further supports that our method learns to use tools effectively and is therefore robust across different tools. **When the tools themselves have comparable capability, how they are used matters more than which tool is chosen.**
>
> >**W2**: Why existing methods still lag behind by ~20% even with GT region crops.
>
> **R2**: Thank you for raising this valuable question. Mere access to region crops is insufficient; what truly matters is the ability to **understand and reason over the regional inputs** returned by tools. Existing MLLMs mainly operate at the image level and are not well aligned with such cropped inputs, which are typically absent from their pretraining and fine-tuning stages. As a result, even with GT region crops, they still struggle to align textual reasoning with region-level visual features, as reflected in Table 3. This explains why their performance remains substantially below ours. Our results therefore support the central contribution of our method: internalizing tool-returned results into the reasoning process, together with strategic training, is critical for eliciting pixel-level visual reasoning ability.
>
> >**W3&Q2**. Computational overhead.
>
> **R3**: Thank you for this valuable suggestion. To evaluate efficiency and computational cost, we sample 1,600 instances from the eight VQA benchmarks used in testing, with 200 randomly selected samples from each dataset, and measure the average inference time per sample of different models on an A100 GPU.
>
> |Model|Qwen2.5-VL-7B|Chiron-o1-8B|LLaVA-Med-7B|DeepEyes-7B|MMedAgent-7B|Ophiuchus-7B|
> |---|---:|---:|---:|---:|---:|---:|
> |Time(s)|3.6|3.9|3.7|4.2|9.3|4.5|
>
> Ophiuchus remains competitive in inference overhead. It is also faster than existing medical agents such as MMedAgent. In real-world deployments, the computational overhead is minimized by modern serving infrastructure, such as microservice-based tool deployment, GPU-resident model serving, KV caching, and Encode-Prefill-Decode (EPD) disaggregation, which can efficiently batch the processing of new image crops during the prefill phase. Moreover, as shown in Section D.6, the model learns to invoke tools efficiently. It selectively invokes tools **only when necessary**, achieving a good balance between accuracy and efficiency.
>
> >**W4&Q4**. Can the framework integrate other tools for pathology examination?
>
> **R4**: Thank you for this valuable suggestion. We fully agree that SAM2 and BioMedParse are limited for pathology tasks. Following your suggestion, we incorporate VISTA-PATH (Liang P, et al.) as an additional pathology-specific segmentation tool and evaluate it on two WSI classification benchmarks, TCGA-BRCA and TCGA-NSCLC. Ophiuchus uses VISTA-PATH for key-region segmentation, followed by zoom-in patch navigation, multi-view sequential reasoning, and final classification. As shown by the balanced accuracy results below, Ophiuchus effectively leverages VISTA-PATH and outperforms several pathology-specific large multimodal models, including Quilt-LLaVA, PathGen-LLaVA, and CPath-Omni.
> |Model|BRCA|NSCLC|
> |---|---:|---:|
> |GPT-4.1|60.7|65.4|
> |GPT-5|61.9|66.8|
> |DeepEyes-7B|53.7|59.1|
> |Lingshu-7B|58.5|56.3|
> |Quilt-LLaVA-v1.5-7B|55.3|54.2|
> |PathGen-LLaVA-13B|66.4|54.1|
> |CPath-Omni-15B|78.5|85.8|
> |Ophiuchus|71.7|80.9|
> |Ophiuchus+VISTA-PATH|87.4|88.7|
>
> >**W5&Q3**. How to validate the constructed dataset?
>
> **R5**: Our data undergoes rigorous verification through both strict LLM-based screening and expert review. As stated in Section 3.3 (Lines 267–269), two radiologists further cleaned the synthetic dataset by removing unreasonable or clinically uninformative samples. Please refer to our response to W1 in Review fYSv for details.

---

> > ### Author Rebuttal · Reviewer_ezX1 · 2026-04-02
> >
> > Thanks the authors for their detailed responses. However, I remain unconvinced by the results, particularly regarding the source of the reported performance gains. The main results presented in the primary table appear somewhat inconsistent with the ablation studies.
> > If the improvements are primarily driven by enhanced visual reasoning, one would expect other MLLMs to achieve comparable performance. Conversely, if automated tool selection plays a critical role, this should be clearly supported by the ablation results. At present, the evidence does not provide strong justification for either of these hypotheses.
> > Therefore, I am inclined to maintain my original score.

---

> > > ### Author Response · Authors · 2026-04-03
> > >
> > > Thank you for your timely and important response. We are pleased to continue the discussion and provide additional evidence.
> > >
> > > >**''If the improvements are primarily driven by enhanced visual reasoning, one would expect other MLLMs to achieve comparable performance.''**.
> > >
> > > We would like to kindly clarify that this is not the case.
> > >
> > > Our work specifically targets active perception through reasoning—the cognitive process by which a model dynamically updates its visual working memory to resolve ambiguity. Unlike standard MLLMs or medical agents that treat tools as disjoint functional offloads, Ophiuchus internalizes the "crop" operation to actively acquire focused visual information during the reasoning process.
> > >
> > > This is different from standard MLLM agents that simply call external APIs. In our framework, visual operations with tools are learned as internal reasoning steps. This allows the model to autonomously perform practical behaviors like searching for small objects or double-checking uncertain regions before answering. We consider this a solid foundation for active perception.
> > >
> > > In contrast, standard MLLM agents do not possess the same visual reasoning capabilities as Ophiuchus. First, they are unable to effectively integrate visual information with textual reasoning. As a result, even if they can use tool APIs, the key limitation is that they cannot reliably determine when to invoke tools or reason jointly with them. This prevents them from accurately predicting tool parameters and obtaining precise crop results. Second, the visual perception and understanding abilities of standard MLLM agents are also insufficient. Therefore, even when provided with ground-truth crop regions, they still cannot achieve comparable performance.
> > >
> > > To empirically substantiate this, we compared Ophiuchus with three "detect-then-answer" baselines on the testing benchmarks. Specifically, **+self-crop bbox** refers to a two-stage pipeline where the model first predicts the target object and then answers based on the crop; **+gt bbox** provides the ground-truth object crop, representing the upper bound of the reviewer's proposal (assuming perfect detection of the relevant object).
> > >
> > > |Model|PathVQA|SLAKE|VQA-RAD|OmniMed|MMMU(H&M)|MedXpertQA|In-House-VQA|
> > > |---|---:|---:|---:|---:|---:|---:|---:|
> > > |Qwen2.5-VL-7B|50.4|66.5|63.4|64.7|56.7|23.5|20.2|
> > > |+self-crop bbox|52.8|67.6|63.7|65.1|58.5|23.7|20.3|
> > > |+gt bbox|58.7|70.8|67.5|67.2|60.1|28.2|31.5|
> > > |Ophiuchus|74.3|83.9|73.6|78.6|76|39.3|50.4|
> > >
> > > The results confirm that simply concatenating high-quality local visual tokens—even assuming perfect detection—is insufficient. The performance gap highlights that Ophiuchus succeeds not merely by "seeing clearly," but by reasoning iteratively, a capability that static visual concatenation cannot replace.

---

### Official Review · Reviewer_9try · 2026-03-04

**Soundness:** 2
**Presentation:** 4
**Significance:** 4
**Originality:** 4
**Overall Recommendation:** 5
**Confidence:** 4

**Summary:**

This paper proposes Ophiuchus, a medical MLLM agent that performs interleaved vision-language reasoning by adaptively invoking external segmentation and zoom tools during chain-of-thought. The training framework consists of three stages: cold-start SFT on GPT-5-generated tool-use trajectories, self-reflection fine-tuning using trajectory pairs that demonstrate self-correction, and agentic RL with fine-grained rewards covering format, correctness, and conditional tool-use bonuses. The system is evaluated on eight VQA benchmarks, two segmentation sets, and a report generation task.

**Compliance With Llm Reviewing Policy:**

Affirmed.

**Key Questions For Authors:**

1. In Table 1, is the in-domain vs. out-of-domain distinction applies to all methods in the table? Is the average performance over all datasets a meaningful metric? I'm aware similar presentation formats are used in existing work, but I'm wondering the authors thoughts
2. Which BLEU score (e.g., BLEU-2, BLEU-4) is used in Tables 12 & 13? Why the results are different from existing papers - did the authors further trained the models?

Overall a nigh quality paper and I'll update my rating after these items are clarified.

**Limitations:**

Yes.

**Strengths And Weaknesses:**

**Strength**
1. Thinking-with-image is a well-motivated method to solve challenging questions in the medical imaging domain. The self-reflection sampling strategy is novel. It provides an elegant way to mine self-corrective behaviors from training dynamics without requiring additional annotation.
2. Empirical results are substantial and convincing. Gains over tool-augmented baselines using the same zoom-in setup (Table 6) isolate the contribution of training over tool availability.
3. Ablations are thorough: each training stage is ablated, reward components are ablated, tool quantity and type are systematically varied, and hyperparameter sensitivity is reported.

**Weaknesses**
1. Table 1, the main comparison, lacks fairness for the main claims. Ophiuchus has access to the tools and is trained on the benchmark dataset. The paper fine-tunes open-source baselines in Table 9, which fixes the fairness issue and can be more prominently featured.
2. The report generation performance in Table 12 doesn't match existing literatures.

---

> ### Author Rebuttal · Authors · 2026-03-31
>
> We sincerely appreciate the reviewer’s recognition of the novelty of our method and the positive assessment that our experiments are substantial and convincing, with thorough ablation studies.
>
> >**W1**: Table 9 fixes the fairness issue and can be more prominently featured.
>
> **R1**: Thank you for this valuable suggestion. Ensuring a fair comparison is our top priority, and we fully agree with your suggestion. We will therefore move Table 9 to the main paper for clearer presentation.
>
> >**Q1**: In Table 1, is the in-domain vs. out-of-domain distinction applies to all methods in the table? Is the average performance over all datasets a meaningful metric?
>
> **A1**: Thank you for raising this important question. The in-domain vs. out-of-domain distinction primarily refers to our Ophiuchus. For the public benchmarks, including PathVQA, SLAKE, VQA-RAD, OmniMed, MMMU (H&M), and MedXpertQA, Ophiuchus is not trained on their training sets. Therefore, we consider the performance on these datasets to better reflect its generalization ability. The average performance is computed as **the mean over seven out-of-domain zero-shot benchmarks**, namely the public datasets plus In-House-VQA, **excluding D_test-VQA (in-domain)**. We have stated this in the caption of Table 1 and will make it clearer in the revised paper.
>
> For the medical MLLMs compared in Table 1, including LLaVA-Med-7B, HuatuoGPT-Vision-34B, Lingshu-7B, Chiron-o1-8B, Med-R1-2B, MMedAgent-7B, MMedAgent-RL-7B, and VILA-M3-40B, the official weights of these models are **fine-tuned** on the training sets of public medical benchmarks such as PathVQA, SLAKE, and VQA-RAD. Therefore, the comparison is in fact more favorable to them, since most of their evaluations are essentially **in-domain**. Our in-domain vs. out-of-domain split in Table 1 is intended to highlight that Ophiuchus can still outperform these models without being trained on those benchmarks, which further underscores the importance of tool-augmented fine-grained reasoning for medical tasks.
>
> For the remaining general-domain MLLM agents in Table 1, such as DeepEyes-7B, Mini-o3-7B-v1, LLaVA-Next-13B, and PixelReasoner-RL-v1-7B, they—like our Ophiuchus—have **not** been fine-tuned using the training sets provided by public medical benchmarks. Therefore, the in-domain versus out-of-domain distinction **also applies to** them. We also recognize that this setting is not fully fair to these general-domain models, as they are affected by the domain gap. Therefore, in Table 9, we **further fine-tune them on our proposed dataset and tool set** for a more controlled comparison. We will present all these points more clearly in the paper.
>
> In practice, MLLMs are trained on massive amounts of data, and public benchmarks are easily accessible, making it difficult to regard them as truly out-of-domain test sets. As a result, evaluating the true generalization ability of MLLMs remains an important challenge. In addition, public benchmarks often involve relatively narrow task formats and therefore cannot fully reflect the complexity of real-world clinical scenarios. For this reason, we believe that evaluation in **more realistic clinical settings** is particularly meaningful, which is why we constructed the In-House-VQA and In-House-Seg datasets as part of our evaluation. Looking ahead, we believe that building a sufficiently large and diverse public benchmark for medical tasks is an important direction for reliably assessing the capabilities of MLLMs.
>
> >**W2&Q2**: Which BLEU score (e.g., BLEU-2, BLEU-4) is used in Tables 12 & 13? Why the results are different from existing papers
>
> **R2**: Thank you for this thoughtful comment. We use the BLEU-4 score for evaluation. For a fair comparison, we follow the official data split, i.e., 70%/10%/20% for the training, validation, and test sets, respectively. This results in 368,960 samples in the training set, 2,991 in the validation set, and 5,159 in the test set. We do not use the training set for fine-tuning, and instead directly evaluate on the test set. We will clarify these details in the revised paper.
>
> In addition, performance at test time can be affected by hardware conditions and deployment settings, such as temperature, top_k, max_new_tokens, and frequency_penalty across different devices. We have also reviewed several recent works [1,2], and the results reported in Tables 12 and 13 remain broadly within a comparable range to existing methods.
>
> **Reference**
>
> [1] Huang Z, et al. Kiut: Knowledge-injected u-transformer for radiology report generation. CVPR 2023.
>
> [2] Gao Q, et al. MARE: Multimodal Analogical Reasoning for Disease Evolution-Aware Radiology Report Generation. AAAI 2026.

---

> > ### Author Rebuttal · Reviewer_9try · 2026-04-01
> >
> > Thanks the authors for the response. I maintain my original rating. A recommendation is to use more recent metrics to evaluate radiology report generation performance in Tables 12 & 13, such as the ReXrank benchmark. The current table uses NLG metrics were slightly out of date, making people from the field hard to contextualize this work. However, this does not change my rating. Thanks for the work.

---

> > > ### Author Response · Authors · 2026-04-04
> > >
> > > Thank you very much for your valuable and thoughtful suggestions. We also sincerely appreciate you for sharing the ReXrank benchmark with us. We will further improve our manuscript accordingly.

---

### Official Review · Reviewer_bKtv · 2026-03-08

**Soundness:** 3
**Presentation:** 3
**Significance:** 3
**Originality:** 2
**Overall Recommendation:** 4
**Confidence:** 3

**Summary:**

This paper presents Ophiuchus, a medical multimodal agent that interleaves reasoning with visual tool use, such as segmentation and zoom-in operations, to support more fine-grained medical understanding. The method combines cold-start supervised fine-tuning, a self-reflection fine-tuning stage, and reinforcement learning for strategic tool use. Experiments on multiple medical VQA and segmentation benchmarks show substantial improvements over strong baselines, suggesting that integrating tool use into the reasoning loop is a promising direction for medical multimodal models.

**Compliance With Llm Reviewing Policy:**

Affirmed.

**Final Justification:**

I recommend acceptance because the paper presents a technically sound and practically meaningful framework for tool-augmented medical reasoning, with convincing empirical improvements.

**Key Questions For Authors:**

1.How are the “early” and “late” checkpoints selected for self-reflection sampling in practice? Are they fixed epochs, adjacent checkpoints, or selected dynamically per sample?

2.If an example changes multiple times across training (e.g., wrong → correct → wrong, or wrong → correct → correct), how is the reflection pair chosen?

3.How sensitive is the reflection data construction to checkpoint choice, training noise, and trajectory pairing heuristics?

4.Why are only early-wrong / late-correct examples used for the reflection stage? Did the authors consider using persistently difficult early-wrong / late-wrong samples for targeted fine-tuning?

5.Can the authors provide an ablation comparing self-reflection sampling against other hard-example mining strategies, to verify that the gain comes from reflection-specific supervision rather than simply re-training on difficult cases?

6.Have the authors considered evaluating the method on a stronger and more recent backbone, such as Qwen3-VL?

**Limitations:**

yes

**Strengths And Weaknesses:**

Strengths:

1.The paper addresses an important and timely problem in medical multimodal reasoning: enabling models to interact with visual tools during reasoning rather than relying only on static global perception.

2.The overall framework is well motivated, and the idea of interleaving reasoning with segmentation/zoom-based visual evidence is intuitive and practically meaningful for medical tasks.

 Weaknesses:

1.The self-reflection fine-tuning stage is not described clearly enough. The paper does not specify how “early” and “late” checkpoints are selected in practice for reflection sampling.

2.It is also unclear how the method handles unstable cases across checkpoints, e.g., when an example is wrong at one checkpoint, correct at an intermediate checkpoint, and wrong again later. This makes the data construction procedure difficult to reproduce.

3.More broadly, the reflection data construction may be sensitive to checkpoint noise and pair-selection heuristics, but this sensitivity is not discussed or analyzed.

4.The paper only uses early-wrong / late-correct examples for reflection fine-tuning. It is unclear why persistently difficult cases (e.g., early-wrong / late-wrong) are excluded, and whether the gain comes specifically from “self-reflection” rather than from targeted re-training on hard examples more generally.

---

> ### Author Rebuttal · Authors · 2026-03-31
>
> We sincerely appreciate the reviewer’s recognition of the significance of our work and the superiority of our methodological framework.
>
> >**W1&Q1**: How are the ''early'' and ''late'' checkpoints selected for self-reflection sampling？
>
> **R1**: Thank you for this thoughtful comment. We select checkpoints based on training dynamics. Epoch 4 is used as the early checkpoint because the loss has already dropped to a relatively low level and its decline begins to slow, indicating that the model has acquired basic tool-augmented reasoning ability while avoiding trivial errors caused by undertraining. Epoch 12, the final epoch, is used as the late checkpoint because training has largely stabilized and the outputs are more reliable, which helps filter out extremely difficult and noisy samples. We then construct self-reflective pairs from cases that are incorrect at epoch 4 but correct at epoch 12, yielding samples of appropriate difficulty.
>
> >**W2&Q2**: How the method handles unstable cases across checkpoints.
>
> **R2**: Thank you for this insightful comment. Although self-reflection sampling (SRS) shares some similarity with hard-example mining, **their goals are fundamentally different**. Hard-example mining aims to place greater emphasis on difficult or erroneous cases to improve future performance on similar examples, whereas SRS is designed to construct self-corrective reasoning trajectories by pairing an early incorrect reasoning process with a later correct one. Therefore, we only select wrong (epoch 4) → correct (epoch 12) cases. Intermediate fluctuations do not affect this criterion, since our objective is not to mine unstable samples, but to **build trajectories** that teach the model to recognize and revise flawed reasoning or tool use. Therefore, SRS is essentially unaffected by unstable cases across checkpoints. We also agree that further analysis and comparison with hard-example mining are important; please see our responses to W3 and W4.
>
> >**W3&Q3**: Sensitivity analysis of checkpoint noise and pair-selection heuristics.
>
> **R3**: Following your suggestion, we conduct additional analyses and summarize the results in the table below. First, we vary the early and late checkpoint choices while keeping the number of self-reflective samples and the rest of the training pipeline unchanged. The performance remains largely stable, indicating that our reflection data construction strategy is robust to checkpoint noise. Second, we examine different trajectory-pairing heuristics by adding more correct → wrong and wrong → wrong pairs on top of the default wrong → correct setting. Consistent with our discussion in W2, **late-wrong trajectories (e.g., early-wrong → late-wrong)**  are **not conducive** to learning self-reflective reasoning patterns and harm performance, so we **exclude them from fine-tuning**.
>
> |Early|Late|Trajectory Pairing Heuristics|VQA-RAD|SLAKE|MedXpertQA|In-House-VQA|
> |-|-|-|-|-|-|-|
> |4|12|wrong→correct *(default)*|73.6|83.9|39.3|50.4|
> |3|11|wrong→correct|73.3|83.6|39.3|50.2|
> |3|10|wrong→correct|73.3|83.5|39.1|50.1|
> |5|11|wrong→correct|73.4|83.8|39.2|50.3|
> |6|10|wrong→correct|73.3|83.7|39.2|50.2|
> |6|8|wrong→correct|73.2|83.4|39.1|50.1|
> |4|12|wrong→correct & correct→wrong|71.5|80.1|35.4|46.2|
> |4|12|wrong→correct & wrong→wrong|71.9|80.4|35.8|46.7|
>
> >**W4&Q4&Q5**: Add comparison against hard-example mining strategies.
>
> **R4&5**: Thank you for this insightful suggestion. We further compare SRS with two classical hard-example mining strategies, namely Online Hard Example Mining (OHEM) [1] and Focal Loss [2], by replacing the SRS strategy used in the targeted fine-tuning stage of Ophiuchus with OHEM and Focal Loss. Specifically, for the OHEM baseline, we prioritize samples with incorrect final answers from the candidate batch in each training iteration. For the Focal Loss baseline, we reformulate the final-answer supervision as a difficulty-aware objective, where correctly answered easy examples are down-weighted. As shown below, the results verify that the gain comes specifically from self-reflection, rather than simply re-training on difficult cases.
> |Methods|VQA-RAD|SLAKE|MedXpertQA|In-House-VQA|
> |-|-|-|-|-|
> |OHEM|72.7|81.4|37.5|47.2|
> |Focal Loss|72.9|81.5|37.7|47.9|
> |SRS|73.6|83.9|39.3|50.4|
>
> >**Q6**: Evaluating the method on Qwen3-VL.
>
> **R6**: Thank you for this constructive suggestion. As shown below, our method consistently improves performance, demonstrating strong generalization across backbones and compatibility with stronger ones.
> |Meth.|VQA-RAD|SLAKE|MedXpertQA|PathVQA|MMMU(H&M)|In-House-VQA|
> |-|-|-|-|-|-|-|
> |Qwen2.5-VL-7B|63.4|66.5|23.5|50.4|56.7|20.2|
> |Qwen3-VL-8B|67.9|71.8|25.7|54.4|60.9|25.8|
> |Ophiuchus(Qwen2.5-VL-7B)|73.6|83.9|39.3|74.3|76.0|50.4|
> |Ophiuchus(Qwen3-VL-8B)|75.2|86.3|40.2|76.7|78.1|51.7|
>
> **Reference**
>
> [1] Training region-based object detectors with online hard example mining. CVPR 2016.
>
> [2] Focal loss for dense object detection. ICCV 2017.

---

> > ### Author Rebuttal · Reviewer_bKtv · 2026-04-02
> >
> > Thank you for the detailed rebuttal. My main concerns about the self-reflection stage are largely addressed.
> > However, after reading the other reviews and rebuttal discussion. I decide to keep my original score.

---

> > > ### Author Response · Authors · 2026-04-04
> > >
> > > Thank you for your constructive suggestions, and we are truly grateful for your positive comments on our work.

---

### Official Review · Reviewer_fYSv · 2026-03-13

**Soundness:** 3
**Presentation:** 4
**Significance:** 3
**Originality:** 2
**Overall Recommendation:** 4
**Confidence:** 3

**Summary:**

The paper proposes Ophiuchus, a tool-augmented multimodal large language model (MLLM) framework for medical image understanding that enables models to iteratively reason over localized visual evidence during inference. Instead of relying solely on global image representations, the approach allows the model to dynamically invoke external tools, such as segmentation models (SAM2, BiomedParse) and zoom-in operations, to inspect fine-grained regions and incorporate the resulting observations into a multimodal chain-of-thought reasoning process. To train this capability, the authors introduce a three-stage training pipeline consisting of cold-start supervised fine-tuning for basic tool use, self-reflection fine-tuning to improve decision revision, and agentic tool reinforcement learning to optimize tool-use policies.

**Compliance With Llm Reviewing Policy:**

Affirmed.

**Final Justification:**

The author has provided enough details for me to raise the score. The proposed method is novel with interesting findings; the tool use itself is insensitive to the result, and the tool use frequency correlates with the type of questions, suggesting that the model is not an imitation.

**Key Questions For Authors:**

See Strengths And Weaknesses.

**Limitations:**

yes

**Strengths And Weaknesses:**

The submission presents a technically sound framework that integrates multimodal reasoning with external visual tools for medical image analysis. The proposed three-stage training pipeline: cold-start supervised fine-tuning, self-reflection fine-tuning, and agentic tool reinforcement learning, is well motivated and aligns with recent developments in agentic reasoning models. The empirical evaluation is reasonably comprehensive, covering multiple medical VQA and segmentation benchmarks, including both both in-domain and out-of-domain settings, and comparisons with several strong baselines help support the effectiveness of the proposed approach. The inclusion of ablation studies analyzing the impact of different training stages and reward components further strengthens the empirical validation. Additionally, the paper is generally well written and structured, with clear figures and descriptions.

However, several aspects of the evaluation and analysis could be strengthened. First, the training data relies heavily on synthetic reasoning trajectories generated by large language models, which may introduce bias or imitation effects. The paper does not sufficiently analyze whether the model truly learns improved reasoning strategies or primarily reproduces patterns from the teacher-generated trajectories. Second, although the method emphasizes tool-augmented reasoning, the paper provides limited analysis of the agent’s tool-use behavior, such as when tools are invoked, whether the tool calls are necessary, or how robust the policy is across different scenarios.

---

> ### Author Rebuttal · Authors · 2026-03-31
>
> We sincerely thank you for recognizing the value of our idea, the completeness of our experiments, and the superiority of our method's performance.
>
> >**W1**: Synthetic reasoning trajectories may introduce bias or imitation effects.
>
> **R1**: Thank you for this important comment. (1) **Regarding the potential bias in synthetic reasoning traces.** We apply rigorous verification after data synthesis. Specifically, GPT-o3 is used to check format compliance, tool usage, tool-choice rationale, and clinical correctness. More importantly, **two experienced radiologists** further review and clean the synthetic reasoning trajectories, removing unreasonable or clinically uninformative entries. Their annotations achieve a **Cohen’s kappa of 0.95**, indicating high agreement. (2) **Regarding the concern about imitation effects**. After RL training, the model exhibited three strong reasoning patterns (Supplementary Material E.1). Among them, two are particularly noteworthy. First, Visual Confirmation: in some cases, the model begins with uncertainty but gradually gains confidence by analyzing tool-returned regional observations and zooming in on relevant details to gather evidence and resolve ambiguities. Second, Hallucination Mitigation: although MLLMs may hallucinate at the early stage of reasoning, tool invocation helps the model attend to visual details, thereby reducing hallucination and producing more accurate diagnostic conclusions. Importantly, these two reasoning patterns are absent from the synthetic reasoning traces used during the SFT stage, as those traces follow deterministic reasoning processes. This indicates that the model is not merely imitating the synthetic reasoning patterns learned during SFT, but is instead **exhibiting newly emerged and more sophisticated reasoning strategies**. Following your suggestion, we have included additional analyses. Specifically, we **quantitatively examine the emergence of different reasoning patterns** by **computing the proportion (%) of each pattern** across multiple test benchmarks. The final results are summarized in the table below. The statistics indicate that the model **actively exhibits these emergent high-level reasoning patterns**.
> |Meth.|D_test-VQA|PathVQA|SLAKE|VQA-RAD|OmniMed|MMMU(H&M)|MedXpertQA|In-House-VQA|
> |-|-|-|-|-|-|-|-|-|
> |Visual-Cue Search|38.5|27.1|33.2|30.5|36.3|34.8|38.1|39.9|
> |Visual Confirmation|36.8|20.9|26.4|21.6|27.8|20.3|30.5|35.2|
> |Hallucination Mitigation|16.4|9.6|8.8|7.5|6.2|7.4|10.7|20.3|
>
> In addition, our ablation results in Table 2 show that RL improves the average performance by 7.0% over the SFT-only model, which further supports this conclusion.
>
> >**W2**: More analysis of the agent’s tool-use behavior, such as when tools are invoked, whether the tool calls are necessary, or how robust the policy is across different scenarios.
>
> **R2**: Thank you for your valuable suggestions. We analyzed the issue of over-calling tools in the appendix (Section D.6). Following your suggestion, we further extended this analysis by examining, across different test scenarios, when tools are invoked and whether such tool calls are necessary. We summarize the results below.
>
> Specifically, for each benchmark, we first identify the set of cases that the base model, Qwen2.5-VL-7B, **cannot** answer correctly without using any tools. We regard tool invocation as **necessary** in these cases. We report the proportion of these cases in which Ophiuchus chooses to use tools and answers correctly.
> |Bench.|Percentage of Necessary Tool-Calling Behavior (%)|
> |-|-|
> |D_test-VQA|94.7%|
> |PathVQA|92.5%|
> |SLAKE|97.1%|
> |VQA-RAD|96.7%|
> |OmniMed|94.6%|
> |MMMU(H&M)|93.0%|
> |MedXpertQA|90.8%|
> |In-House-VQA|90.4%|
>
> Then, we identify the set of cases that the base model Qwen2.5-VL-7B **can** answer correctly without using any tools. We regard tool invocation as **unnecessary** in these cases. We report the proportion of such ''directly answerable'' cases for which Ophiuchus still decides to use tools.
> |Bench.|Percentage of Unnecessary Tool-Calling Behavior (%)|
> |-|-|
> |D_test-VQA|4.8%|
> |PathVQA|1.4%|
> |SLAKE|4.2%|
> |VQA-RAD|0.6%|
> |OmniMed|3.5%|
> |MMMU(H&M)|3.7%|
> |MedXpertQA|3.9%|
> |In-House-VQA|4.6%|
>
> The results show that the model tends to invoke tools when fine-grained information is required, and that the vast majority of such tool calls are necessary. In contrast, for questions that the base model can already answer correctly, our model rarely makes additional tool calls, thereby improving performance while maintaining efficiency. This phenomenon is consistently observed across diverse scenarios on multiple benchmarks, demonstrating the robustness of the learned policy.

---

> > ### Author Rebuttal · Reviewer_fYSv · 2026-04-04
> >
> > I thank the authors for the detailed response. All my questions are addressed. I increased the score.

---

> > > ### Author Response · Authors · 2026-04-04
> > >
> > > Thank you for your valuable suggestions, which has greatly helped us improve the clarity of our paper. We are delighted to have resolved all the issues you raised.

---

### Decision · Program_Chairs · 2026-04-30

**Decision:**

Accept (regular)

**Comment:**

There is a strong consensus among reviewers about the novelty of the framework and the training pipeline, and the strong empirical results. There were initial questions about data bias, limited tool-use analysis, baseline fairness, the exact source of performance improvements, etc. Through rebuttal and follow-up discussions, the authors have addressed most issues while Reviewer ezX1's concerns are only partially addressed. Despite this, all reviewers have recommended weak/strong accept. For this reason, the AC recommends Accept for this submission.